# A galanin-positive population of lumbar spinal cord neurons modulates sexual arousal and copulatory behavior in male mice

Constanze Lenschow [1,2,3] ✉, Ana Rita P. Mendes[1,3], Liliana Ferreira[1], Bertrand Lacoste[1], Hugo Marques[1], Nicolas Gutierrez-Castellanos[1], Camille Quilgars [2], Sandrine S. Bertrand [2] & Susana Q. Lima [1] ✉

During sex, male arousal builds to the ejaculatory threshold, allowing genital sensory input to trigger ejaculation. While copulation and arousal are thought to be brain-regulated, ejaculation is a reflex controlled by a spinal circuit. In this framework, the spinal cord is assumed to be strongly inhibited by descending input until the ejaculatory threshold, playing no role in the regulation of copulatory behavior. However, this remains untested. Here we mapped the spinal circuit controlling the bulbospongiosus muscle, essential for sperm expulsion in mice. Our findings show that bulbospongiosus muscle-motor neurons receive input from galanin-expressing neurons, which integrate genital sensory signals. Stimulating these neurons induces bulbospongiosus activity, but responses vary with spinalization, internal state, and decrease with repeated stimulation. Ablating galanin-positive neurons altered ejaculation latency and copulatory patterns. These results suggest that spinal circuits influence not only ejaculation but also copulation and arousal, challenging the traditional view of spinal control in copulation.

Sexual behavior typically begins with pre-copulatory actions, such as courtship displays, that increase sexual motivation (e.g., erection in males and receptivity in females). In species with internal fertilization copulatory actions, including direct genital stimulation, bring the male to the ejaculatory threshold: at this point penile insertion in the female's reproductive organ triggers ejaculation and sperm ejection[1]. This is usually followed by a variable period of decreased male sexual activity[2–4].

While pre-copulatory and copulatory actions are primarily regulated by the brain, in coordination with the autonomic and sensory-somatic nervous systems[5–7], ejaculation in contrast is a two-step reflex controlled by autonomic and somatic circuits within the spinal cord. During the first step, called emission, sperm and seminal fluids are released and accumulate in the prostatic urethra[8,9]. The second step, expulsion, is a somatic reflex caused by the activation of spinal motor neurons (MNs) and the subsequent contraction of the bulbospongiosus muscle (BSM), a large, striated muscle surrounding the base of the penis[10–13].

How copulatory sequences are orchestrated and how the arrival to the ejaculatory threshold is communicated to the spinal cord allowing genital input to trigger the reflex, remains unresolved. Key insights come from experiments in anesthetized male rats, where stimulation of the penis (via electrical stimulation of the dorsal penile nerve) was shown to elicit ejaculation in spinalized animals, but not intact ones[14,15]. While this result strongly supports the idea that genital

[1]Champalimaud Research, Champalimaud Foundation, Av. Brasília, Doca de Pedrouços, Lisbon, Portugal. [2]INCIA, Université de Bordeaux, CNRS UMR5287, Bordeaux, France. [3]These authors contributed equally: Constanze Lenschow, Ana Rita P. Mendes. ✉e-mail: constanze.lenschow@u-bordeaux.fr; susana.lima@neuro.fchampalimaud.org

stimulation is sufficient to trigger the ejaculatory reflex, it also suggests that descending inhibitory signals to the spinal cord may modulate the impact of incoming sensory input, preventing inadvertent ejaculations[16,17]. This contributes to the view of the brain as both a spinal-reflex inhibitor, and a central organizer of sexual behavior[15,18–20]. In this model, arrival at the ejaculatory threshold may be communicated to the spinal circuitry by the transient interruption of descending inhibition, allowing genital stimulation to activate the ejaculatory reflex[5]. Ablation of the rat spinal interneurons controlling the BMS-MNs (a key element of the circuitry controlling the ejaculatory reflex in rats), fully prevented ejaculation, while leaving other aspects of copulatory behavior intact[21], suggesting a minimal role of the spinal ejaculatory network in the organization of pre-copulatory and copulatory behaviors. However, this model remains incomplete, partly because most studies were performed either in humans, where manipulations are limited, or in rats, using methods with poor anatomical and cellular specificity, and/or low temporal resolution.

Here, we took advantage of the house mouse, a species whose sexual behavior consists of copulatory patterns that resemble human sexual dynamics, in particular with respect to repeated vaginal thrusting preceding ejaculation (in contrast to rat sexual behavior, where ejaculation is dependent on the execution of multiple individual penile insertions, spaced in time, each one with the potential of triggering an ejaculation[4,22,23]). To gain access to the circuitry controlling the ejaculatory reflex, we focused on the pelvic muscle responsible for sperm expulsion, the BSM, as a point of entry into the spinal cord. Employing genetic-based strategies with high specificity (viruses combined with transgenic mice) and high temporal resolution (optogenetics and electrophysiology) we identified the mouse BSM-MNs and their presynaptic partners, a group of galanin-positive (Gal+) interneurons in the lumbar spinal cord. We provide functional evidence that the Gal+ neurons receive genital sensory input and that their stimulation evokes dominant BSM activity, but only after spinalization. Surprisingly, the evoked BSM activity depends on the sexual excitement of the male prior to spinalization, indicating that the spinal cord circuitry controlling BSM activity reflects the animal's internal state. While Gal+ neurons appear to be active prior to ejaculation, genetic ablation of this population significantly impacts the length and structure of copulatory behavior. These results point towards an unexpected and more intricate role of the spinal cord during copulation, extending beyond the simple relay of genital information and the control of the ejaculatory reflex.

## Results

### Characterization of the bulbospongiosus muscle motor neurons (BSM-MNs)

To visualize the MNs involved in sperm expulsion, we targeted the BSM to identify the corresponding BSM-MNs in the spinal cord. Fluorogold (FG), a well-established retrograde tracer[24] was injected into the BSM of adult mice (N = 12 mice, Fig. 1A, B). Serial rostrocaudal reconstruction of the spinal cords revealed a consistent distribution of FG-positive (FG+) BSM-MNs in the dorsomedial part of the ventral horn, near the dorsal gray commissure (Fig. 1B). FG+ somas were observed across several spinal segments, spanning from the lumbar 3 (L3) to the sacral 2 (S2) segments (Supplementary Fig. 1), with the majority of cells located between lumbar 6 (L6) and sacral 1 (S1) segments (Fig. 1C). Immunohistochemical characterization of the FG+ somas revealed that BSM-MNs are classic alpha MNs, expressing choline acetyltransferase (ChAT) and osteopontin, which are alpha MN markers[25]. These neurons also receive sensory afferent input, as indicated by the presence of Vesicular Glutamate Transporter 1 (VGLUT1) positive boutons (Fig. 1D). Nearly all FG+ BSM-MNs expressed ChAT (75 out of 76 FG+ cells quantified) and VGLUT1 (76 out of 76 FG+ cells) and 80% were osteopontin positive (61 out of 76 FG+ cells quantified; Fig. 1E). The soma size of FG+ MNs was similar to that classically reported for alpha MNs (Fig. 1F)[26].

To establish a causal relationship between the activity of BSM-MNs and muscle activity, we employed optogenetic methods to selectively activate these MNs[27] while simultaneously monitoring BSM activity with electromyography (EMG; Fig. 1G–M). We injected a retrograde traveling adeno-associated virus (rAAV), carrying the gene for the light activated channel channelrhodopsin (ChR2; rAAV-CAG-ChR2tdTomato, Addgene[28]), into the BSM of postnatal day 3–6 (P3-P6) mouse pups (Fig. 1G, upper panel), as the efficiency of viral particles to infect motor endplates is known to drastically drop at later postnatal days[29], and animals were raised until sexual maturity (2–3 months of age). To activate the ChR2-expressing somas, a laminectomy was performed to allow blue light illumination of several BSM-MNs-containing spinal segments. To assess the specificity of the light stimulation, activity was recorded from both the BSM and a locomotor hindlimb muscle, the *Tibialis Anterior* (TA) (N = 10 mice, Fig. 1G). BSM EMG potentials were tightly locked to laser stimulation (Fig. 1H BSM trace, Supplementary Fig. 2). The largest amplitudes (Fig. 1I), shorter latencies (Fig. 1K), and lowest light intensities (Fig. 1J), were obtained with illumination above the L6 and S1 segments, in agreement with the BSM-MN rostrocaudal density (Supplementary Fig. 2). No activity was observed in the TA (Fig. 1H, TA trace, Supplementary Fig. 2).

In five animals, we performed single-cell juxtacellular recordings from photoidentified BSM-MNs (N = 7 cells; Fig. 1L, MN trace[30]) alongside BSM EMG recordings. Brief blue light stimulation (10 ms, 15 mW) at different frequencies (upper panel: 5 Hz, middle panel: 10 Hz, lower panel: 20 Hz) reliably elicited short latency action potentials (mean latency to spike: 4.643 ms +/− 0.33 ms, Fig. 1M), followed by BSM EMG potentials (Fig. 1H). Spike and EMG fidelities (calculated as the number of spikes or EMG responses divided by the number of light pulses) remained stable up to 20 Hz (Fig. 1N). Moreover, light evoked BSM-MN activity resulted in characteristic pelvic floor movements that were tightly locked to the laser (Supplementary Movie 1), resembling the movements observed during ejaculation in a sexually behaving male (Supplementary Movie 2)[22]. Together, these results revealed a population of MNs, primarily located in the L6 and S1 spinal segments, whose optogenetic activation induces characteristic EMG potentials in the BSM and ejaculatory-like movements of the pelvic floor.

### Characterization of the BSM-MNs presynaptic partners

We next aimed to identify the presynaptic partners of BSM-MNs. To do so, we started by injecting the BSM of adult mice with pseudorabies virus (PRV, Kaplan strain[31–33]) (Fig. 2A). Similar to the FG and retro-AVV injections, PRV-labeled neurons were primarily found in the dorsomedial part of the ventral horn of the L6 segment (Fig. 2B). In the rat, BSM-MNs have been reported to receive input from galanin-expressing (Gal+) interneurons located around the central canal in the lamina X of the L3/L4 spinal segments[21]. Consistent with these findings, dense PRV labeling was observed around the central canal in the L2/L3 spinal segments that overlapped with post-hoc immunohistochemical labeling of galanin (Fig. 2C), revealing the existence of a similar population of Gal+ cells in the mouse.

To specifically access the Gal+ population, we utilized a mouse line where Cre recombinase is expressed under the control of the galanin promoter (Gal-cre mice[34,35]). Gal-cre mice were crossed with a reporter mouse line carrying the gene for the red fluorescent protein tdTomato[36], resulting in progeny expressing the fluorescent protein in Gal+ neurons (Gal-tdT+ , Fig. 2D). Immunohistochemical staining for galanin in the progeny from the Gal-cre x tdTomato cross confirmed the specificity of this mouse line (Fig. 2D & Supplementary Fig. 3). TdTomato-positive neurons were observed along the rostral caudal axis of the spinal cord, namely in laminae X (around the central canal, N = 10 mice, Fig. 2E), with the highest cell density located in the L2/L3 spinal segments (Fig. 2E). The mouse Gal+ neurons expressed enkephalin, cholecystokinin, gastrin releasing peptide and substance P,

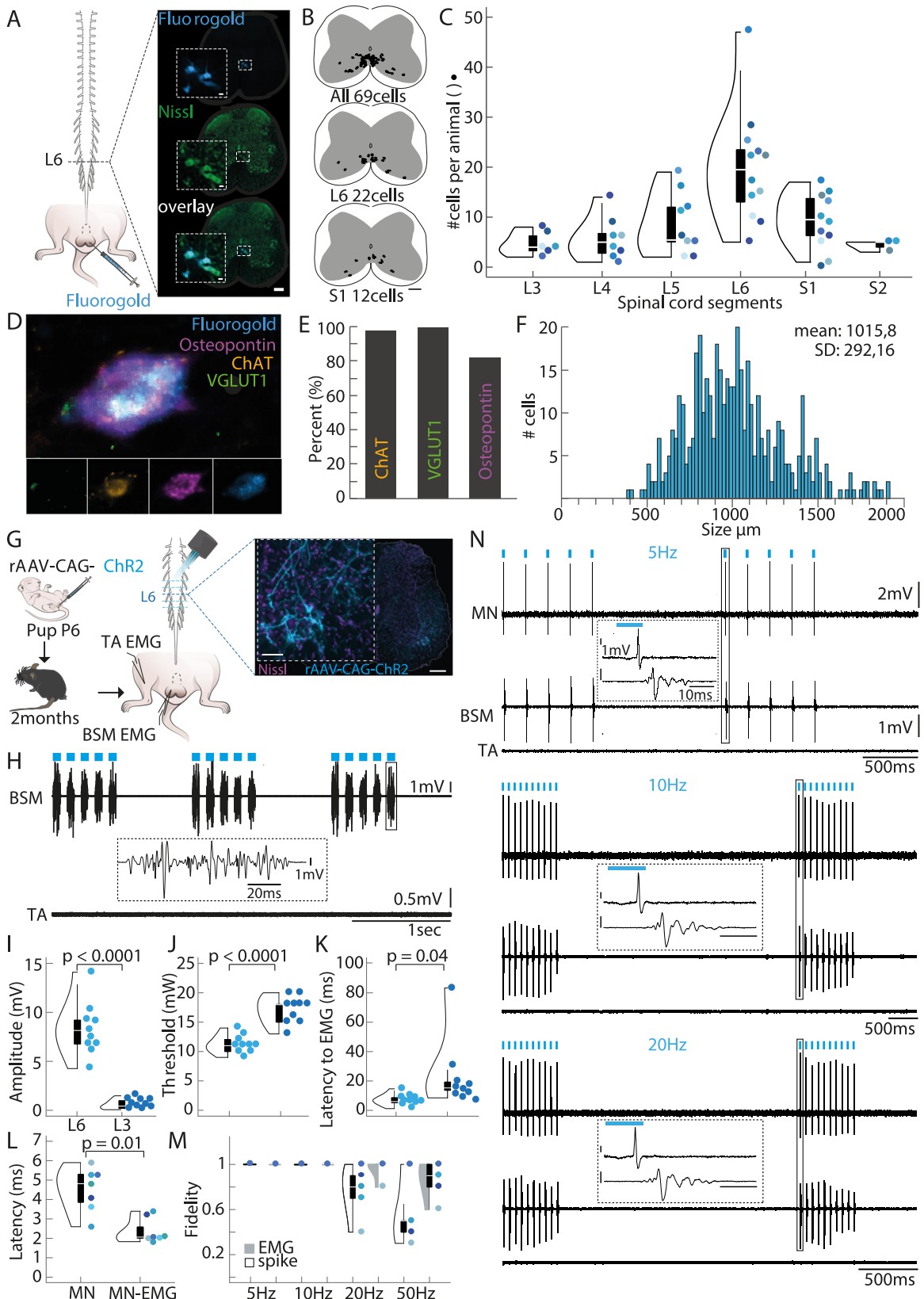

similar to the rat Gal+ neurons[37–39], as all four peptides were present and overlapped with the Gal+ cells surrounding the central canal in the L2/L3 spinal segments (Supplementary Fig. 4).

To determine if the results obtained with the PRV injections are due to the existence of a monosynaptic connection between the Gal+ cells and the BSM-MNs[33], we took a two-pronged approach. First, we

aimed to identify the location of the synaptic terminals of the lumbar Gal+ cells by injecting a Cre-dependent AAV carrying a green fluorescent protein (GFP)-tagged form of synaptophysin (a synaptic vesicle protein, present in neuronal terminals[40]) into the L2/L3 spinal segments of Gal-cre x tdTomato mice (Fig. 2F, N = 7 mice). To visualize the BSM-MNs, mice were simultaneously injected with FG in the BSM

**Fig. 1 | Anatomical and functional characterization of the bulbospongiosus motor neurons (BSM-MNs). A** Left panel: Fluorogold (FG) was injected into the bulbospongiosus muscle (BSM) of adult mice (N = 12), leading to labeling across the rostrocaudal lumbosacral spinal cord. Right panel: FG-positive cells (FG+, blue) within the lumbar spinal segment 6. Nissl stain (green) was used to identify the spinal cord segment based on the atlas[90]. Scale bar 200 μm. Scale bar inset 20 μm. **B** Serial reconstruction of all labeled FG+ cells between lumbar segment 3 and sacral segment 2 revealed their distribution at the dorsomedial ventral spinal cord, close to the gray commissure. Upper panel: distribution of all labeled FG+ cells for one animal; middle panel: cell distribution within lumbar segment 6 (L6); lower panel: cell distribution within sacral segment 1 (S1). **C** Total FG+ cell numbers along the lumbosacral spinal cord. Different colored dots represent different animals (N = 12 mice). Most cells were found at L6 and S1 spinal segments (elements of violin plot: center line, median; box limits, upper (75) and lower (25) quartiles). Animals that lacked cells in a given spinal segment were excluded from the violin plot for that specific segment. **D** Post-hoc immunohistochemical staining for the alpha MN markers osteopontin (purple), choline acetyltransferase (ChAT, orange) and the sensory input marker Vesicular Glutamate Transporter 1 (VGLUT1, green) revealed that all three markers are expressed by FG+ cells (blue) (N = 3 mice). Scale bar 200 μm. Scale bar inset 20 μm. **E** Percentages of FG+ cells expressing ChAT (97.7 ± 2.2%), VGLUT1 (100%) and osteopontin (81.95 ± 2.2%). **F** Cell size distribution of FG+ cells along the rostrocaudal lumbar spinal cord. Frequency histogram depicting number of MNs in each size bin and size of the MNs (binned in 20 μm² steps). **G** Left panel, experimental setup: For functional characterization, rAAV-CAG-ChR2 (ChR2, Channelrhodopsin 2) viral vectors were injected into the BSM of pups (P3-P6), later used for optogenetic stimulation when reaching adulthood. For optogenetic stimulation, a fiber was moved on top and along the rostral caudal lumbar spinal cord while monitoring muscle activity in the BSM and a leg muscle (*Tibialis anterior*, TA) using electromyogram (EMG). Right panel: rAAV-CAG-ChR2 expression (light blue)

in L6 spinal segment (Nissl stain, purple). This experiment was performed in 10 mice. Scale bar 200 μm. Scale bar inset 20 μm. **H** Example BSM and TA EMG trace recorded while optogenetically stimulating above the L6 spinal segment. **I** Higher BSM EMG amplitudes were triggered when illuminating above the L6 spinal segment (mean amplitude 8.37 ± 1.15 mV), compared to illuminating above the L3 segment (mean amplitude 0.57 ± 0.2) (N = 10 mice). Two-tailed Student's t-test p = 0.0000005 (elements of violin plot: center line, median; box limits, upper (75) and lower (25) quartiles). Different colored dots represent different animals (**I–M**). **J** The laser power necessary to elicit BSM potentials was lower above the L6 spinal segment (mean threshold 11.2 ± 0.64 mW), compared to stimulating above the L3 segment (mean threshold 17 ± 0.97 mW) (N = 10 mice). Two-tailed Student's t-test p = 0.00002 (elements of violin plot: center line, median; box limits, upper (75) and lower (25) quartiles). **K** The onset of BSM activity was shorter with illumination above the L6 spinal segment (mean latency to EMG 7.16 ± 1.58 ms) when compared to the L3 segment (mean latency to EMG 23.48 ± 9.43 ms) (N = 10 mice). Two-tailed Student's t-test p = 0.04 (elements of violin plot: center line, median; box limits, upper (75) and lower (25) quartiles). **L** Laser stimulation (10 mW) at 5 Hz (upper panel), 10 Hz (middle panel) and 20 Hz (lower panel) reliably triggered short latency action potentials in a single BSM-MNs. Shortly after the single spikes, BSM EMG potentials were observed (insets) which themselves followed blue laser light applications. No EMG responses were observed in the TA muscle of the leg. **M** Latencies of triggered responses after laser stimulation are plotted for single BSM-MNs (MN; N = 7 mice; mean latency 4.6 ± 0.33 ms) and MN to EMG onset (MN-EMG; mean latency 2.25 ± 0.25 ms). Two-tailed Student's t-test p = 0.001 (elements of violin plot: center line, median; box limits, upper (75) and lower (25) quartiles). **N** Fidelity of spike and EMG activity are shown for the different frequencies of stimulation tested (dark gray: EMG fidelity; black: spike fidelity). Elements of violin plot: center line, median; box limits, upper (75) and lower (25) quartiles (N = 7 cells from 6 different mice).

(Fig. 2F). GFP-labeled terminals belonging to the Gal+ cells (green channel, Fig. 2G) were found around FG+ BSM-MNs (blue channel, Fig. 2G) in all 7 animals (Fig. 2H). On average, a total of 79% ± 2% FG+ labeled cells overlapped with synaptophysin-labeled terminals (Fig. 2H) indicating that the Gal+ neurons in the L2/L3 spinal segments contact the BSM-MNs directly. Furthermore, this approach revealed other projection targets of the Gal+ cells, including the intermediolateral column (IML) and the central autonomic nucleus (CAN) in the lower thoracic and L1 spinal segments (Supplementary Fig. 5), as well as the sacral parasympathetic nucleus (SPN) (Supplementary Fig. 5). The IML and CAN are known to contain sympathetic preganglionic cells that provide sympathetic outflow mainly through the hypogastric and pelvic nerves to the visceral organs[41–44]. The parasympathetic preganglionic cells clustered in the SPN at the S2-S5 spinal segments (Supplementary Fig. 5) innervate the pelvic organs, including the prostate[45], urethra[46,47] and penis[47]. The dorsolateral nucleus (DLN) located at the S1/S2 segments, known to consist of MNs innervating the ischiocavernosus muscle, the most important muscle for erection[48], also contained GFP-labeled terminals of the Gal+ cells. All the regions containing GFP-labeled terminals were also labeled after PRV injection in the BSM (Supplementary Fig. 6), except for the DLN which contains ischiocavernosus MNs. These results suggest that the areas sending information to the Gal+ cells receive reciprocal input, as previously described in the rat[49], and confirm that the PRV initial infection was restricted to the BSM.

To obtain physiological evidence supporting a direct synaptic connection between the Gal+ cells and the BSM-MNs, we performed in vitro whole-cell patch clamp recordings from cholera toxin-B (CTB-594) retrogradely-labeled BSM-MNs in acute spinal cord slices of mouse pups (P2-P6) expressing ChR2 under the galanin promoter (Gal-Chr2, progeny from the cross between the Gal-cre mouse and the ChR2 mouse)[50]. It is important to note that using adult sexually mature animals for this approach is very challenging, as MNs undergo rapid cell death when performing spinal cord slices beyond a certain age, likely due to their complex dendritic branching[51].

An optical fiber (0.2 mm diameter) was placed close to the central canal (Fig. 2I) to illuminate the terminals of the Gal+ cells. A total of 21 CTB-positive BSM-MNs were recorded in Gal-ChR2 pups (N = 7 mice), of which 18 reliably responded to laser illumination (Fig. 2J–M; Supplementary Fig. 7A, B). Short laser pulses led to dominant excitatory postsynaptic potentials (EPSPs) and/or action potentials in CTB-positive BSM-MNs (see upper traces in Fig. 2L, M) also upon consecutive laser application (Supplementary Fig. 7D–F). Repeating laser stimulation after the superfusion of a high cation containing aCSF, to isolate monosynaptic transmission[52], resulted in much smaller EPSPs in BSM-MNs (middle traces in Fig. 2L, M; Supplementary Fig. 7C). Blocking neural transmission via the pharmacological manipulation of NMDA (through AP5) and AMPA (through DNQX) receptors (lower traces in Fig. 2L, M) abolished light-triggered EPSPs in BSM-MNs. In contrast to the BSM-MNs, other large CTB-negative lumbar MNs (N = 16 cells) did not respond to laser illumination (1 out of 16 responded, Fig. 2K, N). Taken together, these findings provide evidence for a population of Gal+ neurons present in the laminae X of the L2/L3 spinal segments in mice, which is monosynaptically connected to BSM-MNs via glutamatergic transmission.

### Sensory input to the Gal+ neurons and the BSM-MNs
It has been hypothesized that once the ejaculatory threshold is reached, genital input can activate the spinal circuitry controlling the ejaculatory reflex[5]. This model is supported by anatomical evidence, as previous studies have reported connections between the sensory branch of the pudendal nerve and putative rat Gal+ neurons[53,54]. However, to the best of our knowledge, functional connectivity has never been established. Therefore, next we investigated whether the BSM-MNs and Gal+ cells receive sensory feedback from the penis.

We first replicated an experiment previously conducted in rats[54,55] (where the penis is stimulated electrically or through air puffs) in male mice that were either intact (Supplementary Fig. 8A) or spinalized (Supplementary Fig. 8C) while monitoring the BSM activity in parallel through EMG. Consistent with findings in rats, we

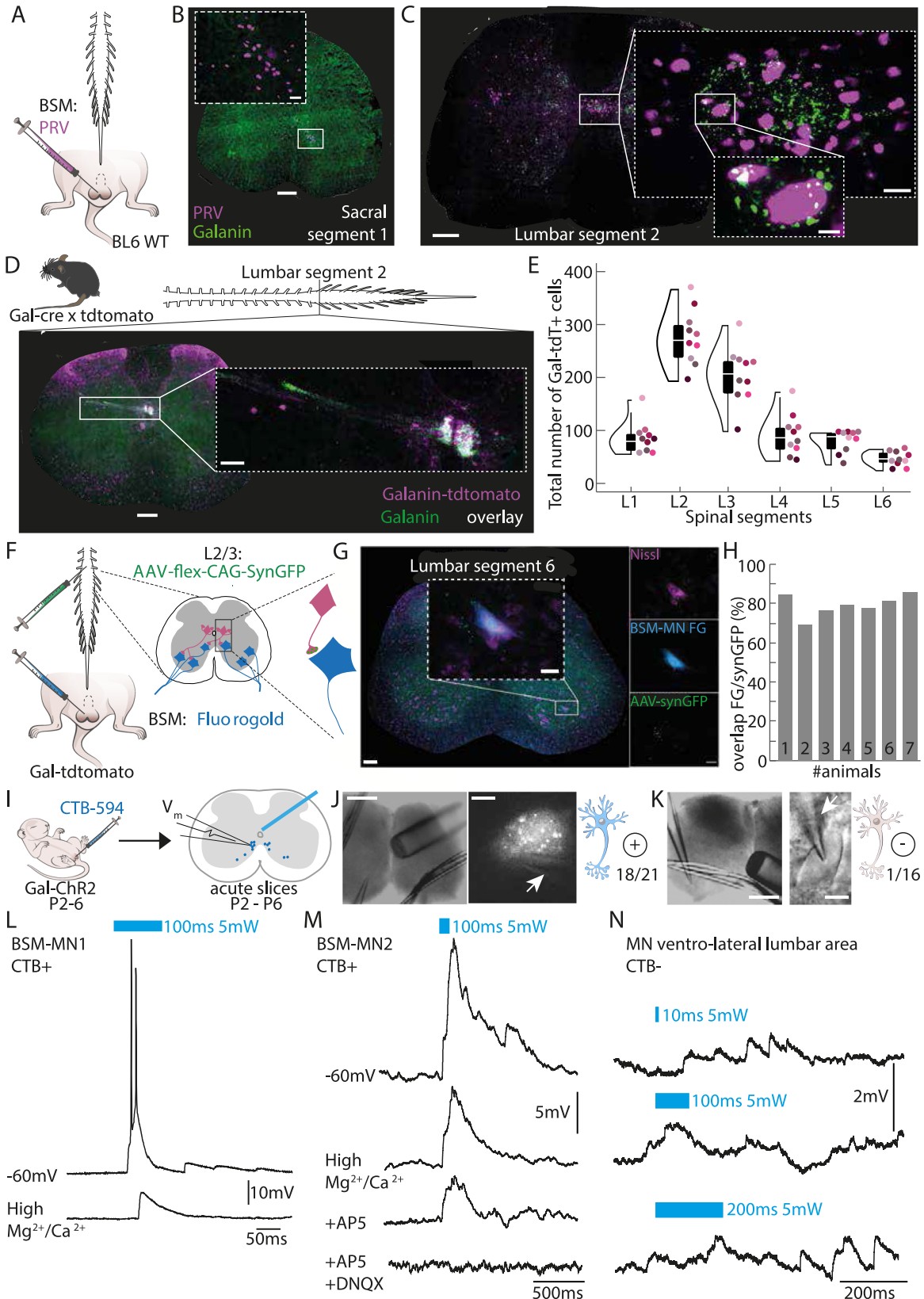

observed prominent BSM EMG activity when stimulating (200 Hz, 100 pulses, 6 V; 5 Hz, 3 × 5 pulses of 100 ms, 6 V) the penis in a spinalized preparation, whereas penile stimulation in intact mice resulted in minimal BSM activation (Supplementary Fig. 8B–D). Penile stimulation-triggered BSM responses were significantly larger in amplitude (Supplementary Fig. 8E) and duration (Supplementary Fig. 8F) in spinalized mice compared to non-spinalized mice, although the onset of BSM responses did not differ (Supplementary Fig. 8G). These experiments further support the existence of supraspinal inhibition of the spinal circuitry, but do not yet confirm whether penile sensory information is physiologically integrated at the level of Gal+ cells and/or BSM-MNs.

**Fig. 2 | The BSM-MNs receive direct input from a group of lumbar spinal cord Galanin positive (Gal+) neurons. A** Experimental setup: pseudorabies virus (PRV) injections were done in the BSM of 6 C57BL6 wild type (BL6 WT) mice. **B** PRV-labeled cells were found at the same major location of FG+ motor neurons (MNs) (see Fig. 1), in the segment L6. Scale bar 200 µm. Scale bar inset 20 µm. **C** PRV injections into the bulbospongiosus muscle (BSM) led to prominent labeling around the central canal at the L2/L3 spinal segments. Post-hoc immunohisto-chemical staining against galanin revealed that the PRV labeled cluster was inter-mingled with galanin-positive (Gal+) immunohistochemical signal surrounding the central canal at the L2/L3 spinal segments (Green: Galanin, Purple: PRV, Scale bar spinal cord section: 200 µm, Scale bar inset 1: 20 µm). **D** Crossing a Gal-cre line with a tdTomato reporter line, and counterstaining the processed spinal cord sections for galanin, revealed a tight overlap between the tdTomato signal (pink) and the galanin signal (green) (Scale bar spinal cord section: 200 µm, Scale bar inset: 20 µm; see also Supplementary Fig. 4). **E** Total cell counts (N = 10 mice) of all Gal-cre x tdTomato positive (Gal-tdT +) cells (y-axis) along the rostral-caudal lumbar spinal cord (x-axis) confirmed the presence of a prominent cluster of Gal-cre x TdTomato cells at the L2/L3 spinal segments (elements of violin plot: center line, median; box limits, upper (75) and lower (25) quartiles). Different colored dots represent dif-ferent animals. **F** Anatomical connection between Gal+ cells and BSM-MNs was investigated through the co-injection of FG into the BSM and a cre-dependent AAV carrying a GFP-tagged synaptophysin (AAV-flex-CAG-SynGFP) into the L2/L3 spinal segments of Gal-cre x tdTomato animals (N = 7 mice). **G** Example image of a spinal cord section obtained from an animal that received FG and AAV-flex-CAG-SynGFP injections. Inset shows a FG+ BSM-MN that co-localizes with GFP-positive post-synaptic boutons. This was consistent for the 7 mice used in this experiment. **H** Quantification of percentage of FG+ BSM-MNs showing co-expression of GFP-

positive postsynaptic terminals for all 7 injected Gal-cre x tdTomato male mice (mean percentage 79.31 ± 2.1%). **I** Experimental setup for establishing the functional connectivity between the Gal+ cells and BSM-MNs; in vitro whole cell recordings from choleratoxin-B (CTB, tagged with 594 fluorophore) retrogradely labeled BSM-MNs in acute spinal cord slices of Gal-Chr2 pups aged P2-P6. **J** Example spinal cord slice showing the location of fiber and pipette placement (left panel) during the recording of a CTB-positive cell (middle panel). A total of 21 CTB-positive cells were recorded, of which 18 were connected to Gal-ChR2+ fibers running through the dorsal gray commissure above the central canal (right panel) (N = 7 mice). Scale bar: 250 µm left, 5 µm right. **K** Example spinal cord slice showing the location of fiber and pipette placement (left panel) during the recording of a control cell, a large MN located at the lateral ventral horn and CTB-negative (middle panel). A total of 16 CTB-negative cells were recorded out of which 1 seemed to be connected to Gal-Chr2+ fibers running through the dorsal gray commissure above the central canal (right panel) (N = 7 mice). Scale bar: 250 µm left, 5 µm right. **L** Example of a whole-cell recording from CTB-positive BSM-MN. Upper trace: a 100 ms laser stimulation led to a short latency Excitatory Postsynaptic Potential (EPSP) and action potentials. Lower trace: High concentration of Mg+ and Ca2+ in the ACSF lowered the amplitude and latency of the light triggered EPSP. **M** Second example of a CTB-positive BSM-MN is shown, same as (**L**). Lower trace: Application of NMDA (AP5, D-2-amino-5-phosphonopentanoate) and AMPA (DNQX, 6,7-dinitroquinoxaline-2,3-dione) receptor blockers abolished the light-evoked EPSP, indicating that the neural transmission between Gal+ cells and BSM-MNs is glutamatergic. **N** Example of a whole-cell recording of a CTB-negative MN. In contrast to the high number of CTB-positive cells responding to laser stimulation, only in 1 out of 16 CTB-negative cells did the stimulation (Upper trace: 50 ms, Middle trace: 100 ms, Lower trace: 200 ms) led to an EPSP.

To address this question, we used an optogenetic approach and mapped light-induced local field potentials (LFP) of ChR2 infected BSM-MNs in the lumbar spinal cord of adult male mice combined with BSM EMG recordings (Fig. 3A). Prominent time-locked LFP deflections were observed when illuminating above the BSM-MNs-containing spinal segments (Fig. 3B, blue trace, Vm) which were fol-lowed by BSM activity (Fig. 3B, black trace, BSM EMG). After mapping the optimal location of laser induced BSM-MN LFP deflections and BSM EMG activity, the position of the glass pipette capturing LFP deflections was maintained while applying brief air puffs of 5 Hz (10 ms, 1 bar) to the pulled-out penis, or the leg as a control (Fig. 3C). Air puff stimulations led to larger LFP deflections when applied to the penis (Fig. 3C, left panel, blue trace) compared to leg puffs (Fig. 3C, middle panel, yellow trace), a result that was consistent at the population level (Fig. 3C, right lower panel). Plotting the light-induced BSM-MNs LFPs against the LFP deflections induced by penis puffs (blue) or the leg puffs (yellow) showed a higher correlation between the light-induced LFPs and the penis puffs (Fig. 3C, right upper panel). These results hint to the BSM-MNs receiving sensory input from the penis.

We repeated the same experiment in male mice expressing ChR2 under the Gal promoter (Gal-ChR2), as we mapped light-induced LFP signals at the location of the Gal+ cells (Fig. 3D) while monitoring BSM activity in a spinalized preparation. Optogenetic stimulation at the L2/L3 spinal segments evoked prominent BSM responses (Fig. 3E, black trace) in line with time-locked LFP deflections (Fig. 3E, pink trace). We then performed air puff stimulations on the penis or leg while maintaining the LFP capturing pipette at the spot of the most effective light-induced LFPs (Fig. 3D, right panel). Penile sensory stimulation elicited significantly larger LFP responses (Fig. 3F, left panel, pink trace) compared to leg air puffs (Fig. 3F, middle panel, yellow trace and right lower panel). Moreover, light-induced LFPs were more correlated with LFPs induced by penis puffs than those elicited by leg puffs (Fig. 3F, right upper panel) indicating that penile sensory inputs can reach the Gal+ cells present at the L2/L3 spinal segments. These findings provide evidence that both the BSM-MNs and the Gal+ population are the recipients of penile sensory information.

## Description of electrically induced BSM activity

To further explore the role of the Gal+ neurons in the control of the BSM activity in mice, we investigated whether the artificial activation at their location could trigger BSM activity. First, we performed electrical stimulations along the rostrocaudal lumbar spinal cord by inserting a tungsten electrode at various depths (550−850 µm), while con-currently measuring BSM activity using EMG recordings (Fig. 4A) in sexually naive, spinalized animals (SN, N = 8 mice). We simultaneously monitored the activity of the TA leg muscle to assess the specificity of our protocol (Fig. 4A).

The site with the strongest electrically triggered BSM responses was labeled by electrolytic lesions and immunohistochemical staining for galanin (Fig. 4B, upper panel). Successful BSM muscle activity was eli-cited (Fig. 4B and Supplementary Fig. 9) when stimulations (40 µA, 200 Hz, 100 pulses) were applied near the Gal+ cluster, located in the L2/L3 spinal segments (Fig. 4C), at a depth of 850 µm, which aligns with the location of the central canal (Fig. 4D and Supplementary Fig. 9C,D). The electrically triggered BSM responses were significantly higher and longer in spinalized preparations compared to non-spinalized ones (Supple-mentary Fig. 10), suggesting descending inhibition from the brain. This finding aligns with our previous results from the mechanical/electrical stimulation of the penis experiments (Fig. 3) and prior studies in rats[16,17].

However, unlike what was observed in the rat, where repeated electrical stimulations led to consistent BSM responses[56], repeated stimulations in mice resulted in decreased BSM responses (see "Methods"; Fig. 4B; BSM 2nd and 3rd). Quantification of the EMG responses (Fig. 4E, F) showed significantly larger and longer potentials during the first train of stimulation (mean amplitude 0.99 mV +/− 3.6 mV, 1st vs 2nd amplitude p = 0.04, 1st vs. 3rd amplitude p = 0.02; mean length 8.83 ms +/− 1.4 ms; 1st vs 2nd length p = 0.09, 1st vs. 3rd length p = 0.02; Mann-Whitney-U Test) compared to the second (mean amplitude 0.5 mV +/− 0.25 mV; mean length 4.22 s +/− 1.47 s) and third trains (mean amplitude 0.06 mV +/− 0.04 mV; mean length 1.74 s +/− 1.31 s) of current application, while the onset of the responses did not change (mean onset of 1st 3.9 +/− 1.83 s, mean onset of 2nd 6.03 +/− 3.02 s, mean onset 3rd 0.54 +/− 0.13 s; Fig. 4G). The reduction in response during repeated stimulation was not due to deterioration of the preparation (Supplementary Fig. 11).

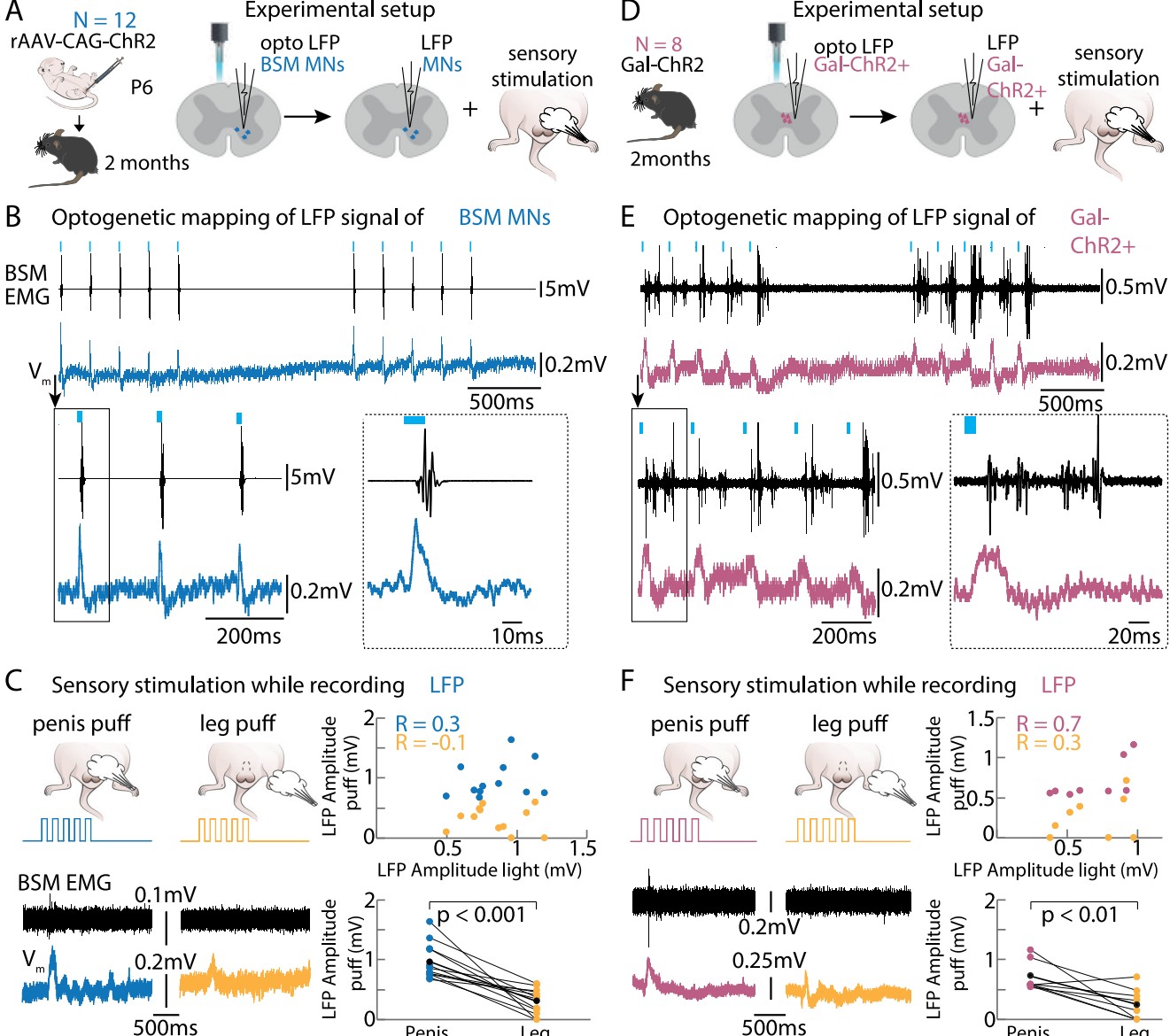

**Fig. 3 | Gal+ neurons and BSM-MNs receive sensory input from the penis.**
**A** Light-induced local field potentials (LFPs) were mapped in adult male mice whose bulbospongiosus muscle motor neurons (BSM-MNs) were infected with Channelrhodopsin 2 (ChR2) at young age (rAAV-CAG-ChR2 injections into the BSM at postnatal day P3-P6). Once the position with the most prominent light triggered LFPs was detected, the glass pipette was left at that position and sensory puff stimulations of penis and leg were conducted. **B** Example traces for optogenetically induced LFP activity of BSM-MNs. BSM (black) and LFP responses (blue) were tightly locked to the laser onset. (EMG: electromyogram). **C** Left panels: The pipette was kept at the position where the highest LFP responses were encountered. Subsequently sensory air puff stimulation of penis (left, dark blue) and leg (right, yellow) were conducted while monitoring LFP and EMG responses in animals whose BSM-MNs were infected with ChR2. Example traces were obtained from the same animal and correspond to the LFP traces depicted in A2. Right upper panel: The highest light-induced LFP responses are plotted against the penis/leg puff induced LFP responses revealing that penis puff-induced LFPs (blue) are more strongly correlated with the light-induced LFP than the leg puff-induced LFP responses (yellow). Every dot is the pooled data of an animal (N = 12 mice). Right lower panels: Penis puff responses (blue) led to significantly higher amplitudes (mean

0.96 ± 0.27 mV) in LFP than leg puff responses (yellow; mean amplitude 0.31 ± 0.09 mV). Two-tailed Student's t-test p = 0.00014. **D** Similarly to (**A**), light-induced LFPs were first mapped in Gal-ChR2 mice. Afterwards, the mapping pipette was left at the location where the strongest light triggered LFPs were encountered. Subsequent sensory stimulation of penis and leg were conducted while monitoring LFP and EMG activity in parallel. **E** Example traces for optogenetically induced LFP activity in Gal-ChR2 animals (pink) while BSM activity (black) was monitored in parallel. Note the light-locked responses of the BSM with parallel timed LFP activity (putative Gal+ population activity around the central canal in L2/L3). **F** Same as (**C**) but for Gal-ChR2 animals. Left panel: Example traces for penis (pink) and leg (yellow) puff induced LFPs correspond to the traces shown in (**E**). Right upper panel: Highest light-induced LFP responses are plotted against the penis/leg puff induced LFP responses revealing that penis puff induced LFPs (pink) are more strongly correlated with the light-induced LFP than leg puff induced LFP responses (yellow). Every dot is the pooled data of an animal (N = 8 mice). Right lower panel: Penis puff responses (pink) led to significantly bigger LFPs (mean amplitude 0.73 ± 0.09 mV) than leg puffs (yellow; mean amplitude 0.25 ± 0.09 mV). Two-tailed Student's t-test p = 0.0035.

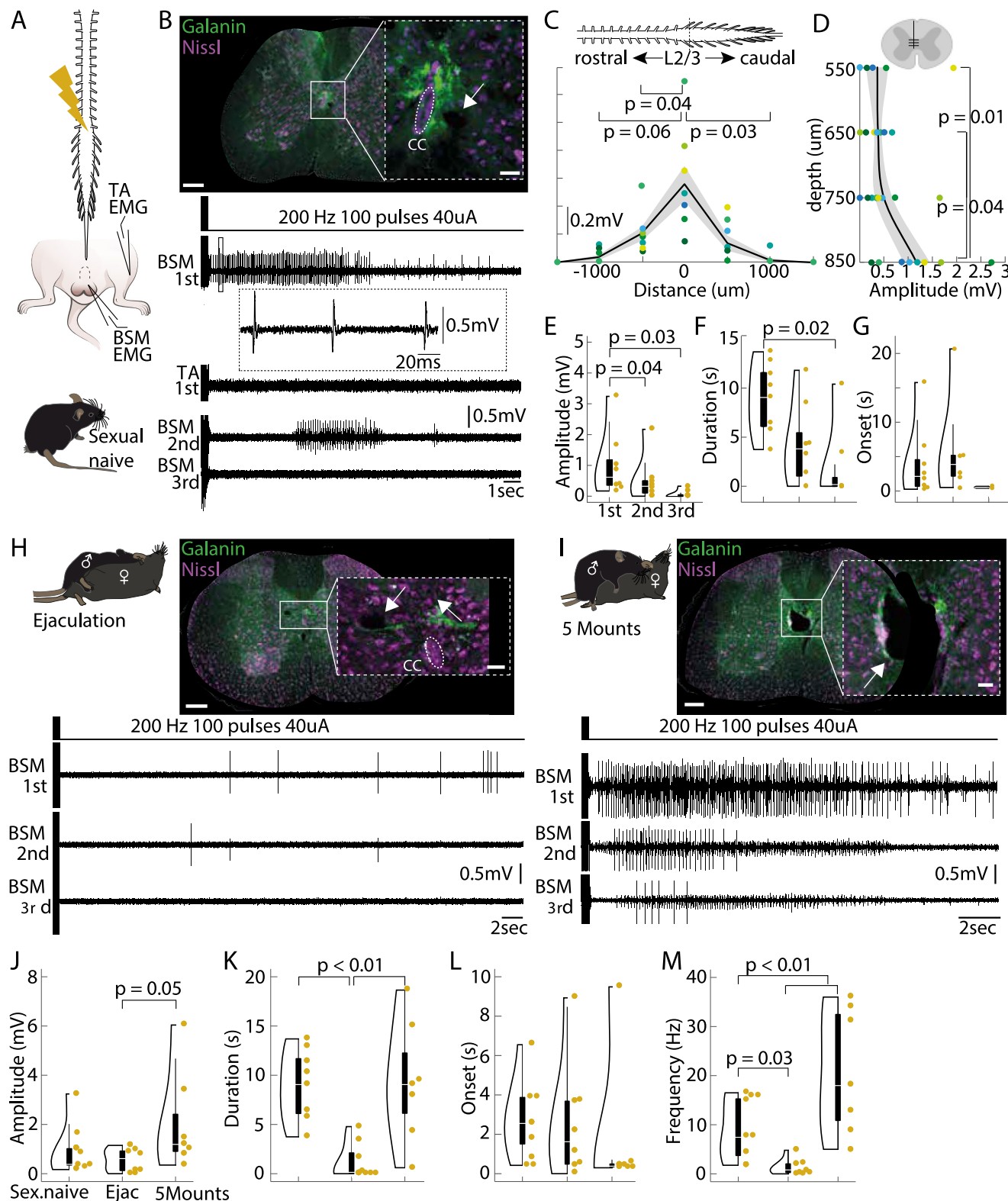

In addition to the intriguing depressing response observed with repeated electrical stimulations, the structure of the electrically elicited BSM activity in mice differed from previously recorded EMG responses during ejaculation in anaesthetized rats (see ref. 56). Moreover, we did not observe the expulsion of sperm or seminal fluid in any of our mouse experiments. To rule out the hypothesis that the lack of sperm emission in mice was due to a technical issue, we replicated the experiment by Borgdorff et al. (2008)[56] in anesthetized rats.

As expected, the application of electrical current (Supplementary Fig. 12A) at the L3/L4 spinal segments (as confirmed by electrolytic lesions; Supplementary Fig. 12B) resulted in the characteristic activity pattern in the rat BSM ($N = 5$ rats; Supplementary Fig. 12C) and the expulsion of sperm (Supplementary Fig. 12D).

To explore how electrically induced BSM activity relates to natural physiological conditions, we conducted in vivo EMG recordings of BSM activity in sexually behaving mice (Supplementary Fig. 13A). BSM

**Fig. 4 | Electrical stimulation of the lumbar spinal cord location harboring the Gal+ neurons leads to BSM activity in an anesthetized preparation. A** Electrical stimulations (200 Hz, 100 pulses, 40 μA) were performed along the rostrocaudal lumbar spinal cord in adult anesthetized and spinalized C57BL6 mice (left panel) while performing electromyogram (EMG) recordings in the bulbospongiosus muscle (BSM) and a control leg muscle (TA - *Tibialis anterior*). **B** Electrolytic lesions were placed at the location where electrical current applications led to the most prominent BSM potentials (upper panel, see inset and white arrow; Green: immunohistochemical staining for Galanin, Purple: Nissl stain). Lower panel: representative traces of the EMG activity (BSM and TA) during electrical current application at the lesion site. While the first current application led to high amplitude and high frequency discharges in the BSM (but not in the leg muscle), second current applications led to a reduced response (in this example, BSM activity was not observed during the third current application). This was consistent for the 8 animals tested in this experiment. Scale bar 200 μm. Scale bar inset 20 μm. **C** Diagram showing the triggered BSM activity along the rostrocaudal spinal cord axis. Largest BSM responses were encountered at the L2/L3 spinal segments (mean amplitude 0.56 ± 0.13 mV). Different colored dots represent different animals (*N* = 8 mice). Mean amplitudes: at 500 μm rostral to L2/L3 0.2 ± 0.05 mV; 500 μm caudal to L2/L3 0.14 ± 0.05 mV; at 1000 μm rostral to L2/L3 0.04 ± 0.02 mV; at 1000 μm caudal to L2/L3 0.01 ± 0.003 mV; represented as mean amplitude values +/− SEM. *P*-values result from a two-tailed Mann-Whitney-U Test. **D** BSM amplitudes plotted against the depth of the stimulation sites. Differently colored dots refer to individual animals (*N* = 8 mice). Mean amplitudes: at 550 μm 0.36 ± 0.23 mV, at 650 μm 0.37 ± 0.083; at 750 μm 0.49 ± 0.18 mV; at 850 μm 1.17 ± 0.27 mV; represented as mean amplitude values +/− SEM. *P*-values result from a two-tailed Mann-Whitney-U Test. **E** Violin plot illustrating the amplitudes of electrically triggered BSM activity (*N* = 8 mice) during the 1st (mean amplitude 0.99 ± 0.36 mV), 2nd (mean amplitude 0.5 ± 0.25 mV) and 3rd (mean amplitude 0.06 ± 0.04 mV) rounds of current application (elements of violin plot: center line, median; box limits, upper (75) and lower (25) quartiles). *P*-values result from a two-tailed Mann-Whitney-U Test. **F** Same as (**E**), but the duration of the BSM activity is plotted. Mean durations: for 1st stimulation 8.8 ± 1.3 s; after 2nd current application 4.2 ± 1.46 s; after 3rd stimulation 1.74 ± 1.3 s. *P*-values result from a two-tailed Mann-Whitney-U Test. Elements of violin plot: center line, median; box limits, upper (75) and lower (25) quartiles. **G** Same as (**E**), but the onset with which BSM activity was triggered is plotted. Mean

onset of EMG: after 1st current application 3.99 ± 1.83 ms; after 2nd stimulation 6.02 ± 3.01 ms; after 3rd current application round 0.54 ± 0.13 ms. A nonparametric ANOVA, two-tailed Wilcoxon-Signed-Rank Test, led to no significance (*P* = 0.21; Wilcoxon-Signed-Rank Test). Elements of violin plot: center line, median; box limits, upper (75) and lower (25) quartiles. **H** Example BSM EMG traces of 1st, 2nd and 3rd current applications, recorded from an animal that ejaculated prior to the electrical stimulation experiment. Note that the BSM activity pattern during 1st and 2nd round of current application is markedly different from the sexually naive animal (**B**). This was consistent for the 8 animals tested in this experiment. Scale bar 200 μm. Scale bar inset 20 μm. **I** Same as (**H**), but traces are obtained from an animal that was allowed to perform 5 mounts with vaginal thrusting prior to the electrical stimulation experiment. BSM activity is comparable to the sexually naive male (panel B). This was consistent for the 7 animals tested in this experiment. Scale bar 200 μm. Scale bar inset 20 μm. **J** Violin plot illustrating the amplitudes of electrically triggered BSM activity in sexually naive males (*N* = 8 mice, mean amplitude 0.99 ± 0.36 mV), males that reached ejaculation (Ejac, *N* = 8 mice, mean amplitude 0.57 ± 0.16 mV) or executed 5 mounts with vaginal thrusting (*N* = 7 mice, mean amplitude 2.03 ± 0.76 mV). Data refers to the mean amplitude on the stimulation site with the highest response during first rounds of current applications. *P*-values result from a two-tailed Mann-Whitney-U Test. Elements of violin plot: center line, median; box limits, upper (75) and lower (25) quartiles. **K** Same as (**J**), but the duration of the BSM activity is plotted. Mean duration: in sexually naive animals 8.8 ± 1.3 s; in the Ejac group 1.26 + 0.26 s; in the group with 5 mounts 10.43 + 1.82 s. *P*-values result from a two-tailed Mann-Whitney-U Test (Ejac vs Sexual Naive: 0,000205136; Ejac vs 5 Mounts: 0,00024219; 5 Mounts vs Sexual Naive: 0,500171095). Elements of violin plot: center line, median; box limits, upper (75) and lower (25) quartiles. **L** Same as (**J**), but the onset of BSM activity is depicted. Mean onset: sexually naive 3.99 ± 1.83 ms; Ejac 2.56 ± 1.04 ms; 5 mounts 0.88 ± 0.48 ms. No significance was found using a non-parametric ANOVA, twotailed Wilcoxon-Signed-Rank Test; *P* = 0.2. Elements of violin plot: center line, median; box limits, upper (75) and lower (25) quartiles. **M** Same as (**J**), but the frequency of BSM activity is shown. Mean frequency: sexually naive 8.79 ± 2.2 Hz; Ejac 1.39 ± 0.56 Hz; 5 mounts 20.81 ± 4.8 Hz). *P*-values result from a two-tailed Mann-Whitney-U Test (Ejac vs Sexual Naive: 0,005777393; Ejac vs 5 Mounts: 0,000853716; 5 Mounts vs Ejac 0,033475703). Elements of violin plot: center line, median; box limits, upper (75) and lower (25) quartiles.

---

activity was detected at various stages during the sexual interaction (Supplementary Fig. 13B). Notably, distinct BSM contractions were observed with each male thrust (Supplementary Fig. 13C), with the strongest bursts occurring as the penis exited the vagina. In contrast, no BSM contractions were present during probing behavior (the period after mounting and before penile intromission, when the male is performing shallow thrusts, trying to locate the female's vagina, Supplementary Fig. 13D). These findings support the hypothesis that the BSM might be involved in maintaining an erection (as the BSM is active during intravaginal thrusts) but not in initiating it.

Interestingly, no BSM activity was detected during the shuddering phase, the brief period immediately following the last thrust leading to ejaculation, characterized by rapid pelvic movements and lasting less than 3 seconds[57] (Supplementary Fig. 13B). Since ejaculation consists of two stages—emission and expulsion—we hypothesize that the shuddering phase corresponds to the emission phase, during which sperm is deposited in the urethra while the penis is less erect[58]. The subsequent expulsion of sperm is driven by BSM contractions observed immediately after the shuddering phase (Supplementary Fig. 13B, E). This hypothesis, suggesting that sperm emission occurs during shuddering and expulsion afterward, is supported by previous studies in mice[57], which found that separating males from females during this phase prevented pregnancy and led to sperm expulsion at the tip of the penis outside the vagina.

As expected, the BSM activity recorded immediately after the shuddering phase showed the highest EMG activity, with bursts averaging a mean amplitude of 0.732 mV, lasting more than 6 seconds (Supplementary Fig. 13F). Based on the same separation studies in mice[57] sperm expulsion is thought to occur during the first 1–2 s

immediately after the shuddering phase occurs. Therefore, after sperm expulsion, the post-shuddering phase activity is probably related to flipping and cupping movements of the penis[57]. Importantly, the dynamics of BSM EMG activity in freely moving animals mirrored those elicited by direct optogenetic stimulation of the BSM-MNs (Fig. 1H) and by sensory stimulation of the penis (Supplementary Fig. 8D). Similar patterns were also observed during electrical stimulation of the location containing Gal+ neurons, including comparable amplitudes and durations, and oscillatory patterns (Supplementary Fig. 13G). These consistent results across experiments further support that we are observing expulsion-like patterns in the BSM, but no sperm emission, when artificially activating the Gal+ cells in anesthetized in vivo preparations.

As previously noted, mice and rats display distinct reproductive strategies, particularly in the number of ejaculations each species can achieve in a short period of time: while rats can ejaculate 7–8 times in a short period before reaching sexual exhaustion, mice, similar to humans, generally enter a refractory period after a single ejaculation, which in the case of C57BL6 mice (the genetic background used in this study) can last several days[4]. We speculated that the inability to elicit stable BSM activity with repeated electrical stimulations in mice might be due to the first round of stimulation inducing a refractory state similar to that following ejaculation.

To test this hypothesis, we allowed male mice to have sex prior to the electrical stimulation experiments. One group of males was allowed to reach ejaculation (Ejaculation, *N* = 8 mice; Fig. 4H upper panel), while another group was allowed to perform 5 mounts with intromission and intravaginal thrusting (5 Mounts, *N* = 7 mice, Fig. 4I upper panel). Electrical stimulation of the spinal cord in the Ejaculation

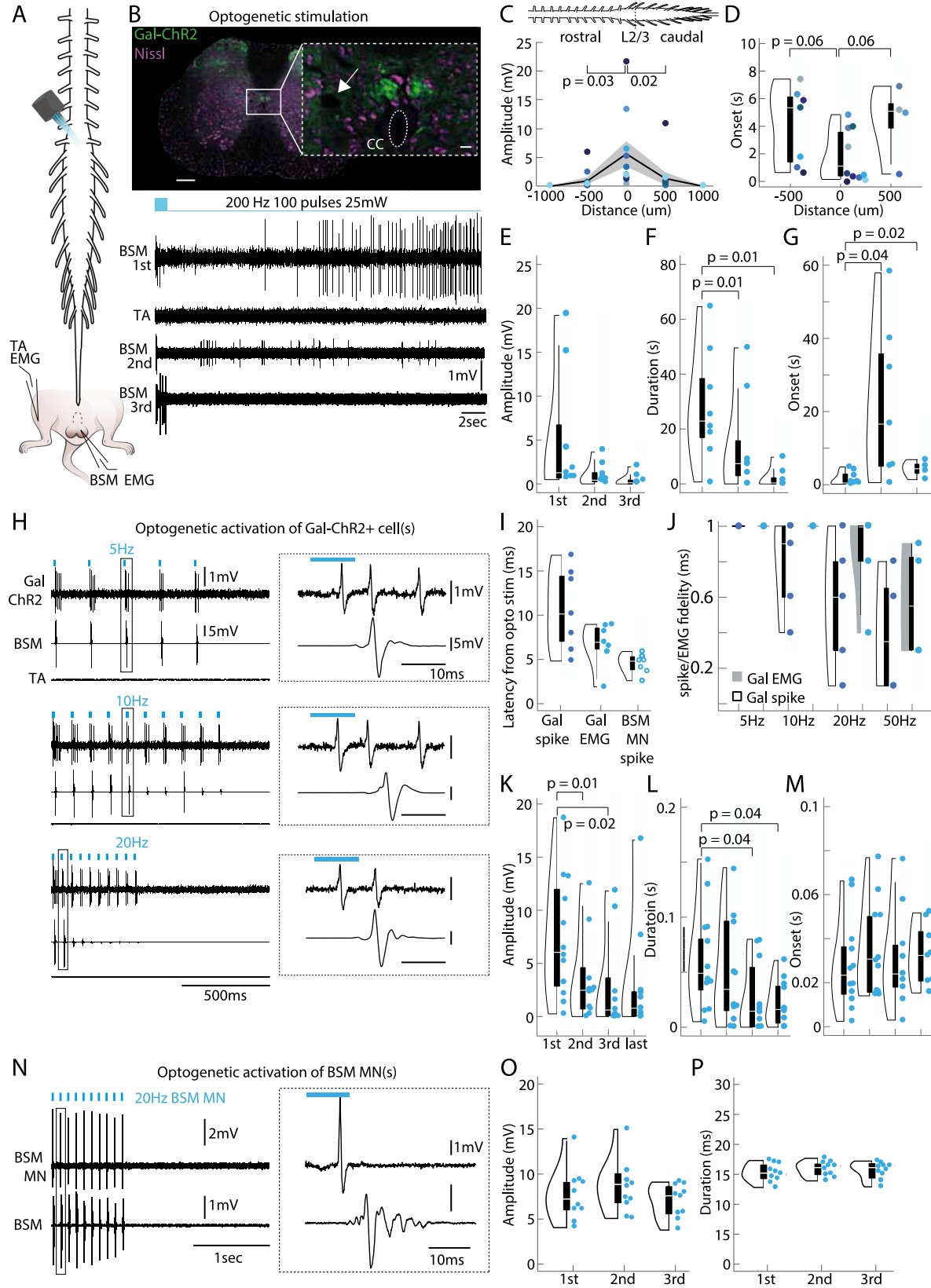

group evoked less BSM activity (Fig. 4H, lower panel) compared to the Sexually Naive group (Fig. 4B, first train of stimulation), and the 5 Mounts group (Fig. 4I, lower panel). The mean amplitude of the BSM activity was significantly higher in the 5 Mounts group (Fig. 4J), while the duration of the BSM activity was significantly shorter in the Ejaculation group compared to the other two groups (Fig. 4K). Although

there was no difference in the mean amplitude of the BSM events between the Sexually Naive and Ejaculation groups (Fig. 4J), the number of events was significantly lower in the Ejaculation group compared to the other two (Fig. 4M), further suggesting that after ejaculation, electrical stimulation of the spinal cord cannot elicit high BSM activity.

**Fig. 5 | Optogenetic activation of the Gal+ neurons leads to BSM activity in an anesthetized preparation. A** Optogenetic stimulations (200 Hz, 100 pulses/10 ms, 20 mW) were performed on top and along the rostrocaudal lumbar spinal cord in adult anesthetized and spinalized Gal-Chr2 males (animals expressing Channer-rhodopsin 2 in Galanin-positive cells, left panel) while performing electromyogram (EMG) recordings in the bulbospongiosus muscle (BSM) and a leg muscle (TA - *Tibialis anterior*). **B** Electrolytic lesions were placed at the location where optogenetic stimulation led to the most prominent BSM potentials (upper panel, see inset and white arrow; Green: Galanin-ChR2, Purple: Nissl stain). Scale bar of spinal cord section 200 μm, inset, 20 μm. Lower panel: representative traces of the EMG activity (BSM and TA) during optogenetic application on top of the lesion site. Note that it led to a similar activity pattern compared to the electrical stimulations: while the first laser application led to high amplitude and high frequency discharges in the BSM, but not in the leg muscle (TA), the second round of light delivery led to a reduced response, and BSM activity was not detected during the third round of optogenetic stimulation. This was consistent for the 10 animals tested in this experiment. **C** Diagram showing the light-triggered BSM activity along the rostrocaudal spinal cord axis. Largest BSM responses were encountered at the L2/L3 segments (mean amplitude 5.61 ± 2.31 mV, *N* = 10 mice). Note that responses are more restricted to the L2/L3 spinal segments where the cluster of Gal+ cells was found, around the central canal (mean amplitudes: 500 μm rostral to L2/L3 0.99 ± 0.58 mV, 500 μm caudal to L2/L3 1.35 ± 1.06 mV; represented as mean amplitude values +/− SEM). *P*-values result from a two-tailed Mann-Whitney-U Test. Different colored dots represent different animals (**C**, **D**). **D** Violin plots with boxplots illustrating the latencies with which BSM responses were triggered as a function of distance (*N* = 10 mice). Shorter latencies were achieved at 0 μm which corresponds to the L2/L3 spinal segments (0.02 ± 0.006 s). *P*-values result from a two-tailed Mann-Whitney-U Test. In 7 out of 10 animals BSM responses were triggered at 500 μm rostral to L2/L3 whereas only in 4 animals BSM responses were triggered at 500 μm caudal to L2/L3 (see individual dots). Elements of violin plot: center line, median; box limits, upper (75) and lower (25) quartiles. **E** Violin plots illustrating the amplitudes of optogenetically triggered BSM signal during 1st (mean amplitude 5.5 ± 2.6 mV), 2nd (mean amplitude 1.48 ± 0.86 mV) and 3rd rounds (mean amplitude 0.41 ± 0.23 mV) of laser application, with boxplots (*N* = 8 mice). *P*-values result from a two-tailed Mann-Whitney-U Test. Every dot represents the BSM responses obtained from an animal. Elements of violin plot: center line, median; box limits, upper (75) and lower (25) quartiles. **F** Same as (**E**), but the duration of BSM activity is plotted. Mean durations: after 1st 29.33 ± 7.28 s; 2nd 15.59 ± 6.43 s; and 3rd 2.27 ± 1.24 s, laser application. *P*-values result from a two-tailed Mann-Whitney-U Test. Elements of violin plot: center line, median; box limits, upper (75) and lower (25) quartiles. **G** Same as (**F**), but the onsets with which BSM activity was initiated are plotted. Mean onset of BSM EMG: after 1st 1.18 ± 0.02 s; 2nd 25.25 ± 0.1; and 3rd 4.28 ± 0.1 s, laser application. *P*-values result from a two-tailed Mann-Whitney-U Test. Elements of violin plot: center line, median; box limits, upper (75) and lower (25) quartiles. **H** In addition to tonic BSM responses as observed during the electrical stimulations, optogenetic stimulations also led to timely locked BSM responses, capable of following stimulation frequencies up to

20 Hz (*N* = 12 mice). In a subset of animals (4 out of 12) we recorded photo-identified single Gal-ChR2 neurons (*N* = 7 cells). Upper left panel: Juxtacellular recording of a Gal-ChR2 cell in parallel with EMG recordings of the BSM (2nd trace) and the TA muscle of the leg (3rd trace) during a 5 Hz laser stimulation. Upper right panel: zoom in of the indicated box illustrating the spike and EMG onset. Middle panels: same as upper panel but for a 10 Hz and 20 Hz (pulses of 10 ms length) laser application of the same cell shown in the upper panel. **I** Violin plot with boxplots illustrating the latency for optogenetically triggered spikes in Gal-ChR2 single cells (Gal spike), the latency between spike onset of Gal-ChR2 single cells and BSM EMG onset (Gal EMG), and the latency to spike of single photo-identified BSM motor neurons (BSM-MNs) (BSM-MN spike). Data refers to 7 single Galanin cells obtained from 4 Gal-ChR2 animals and to 7 single BSM-MNs obtained from 6 animals that received retro AAV injections with ChR2 as pups (see Fig. 1M). Elements of violin plot: center line, median; box limits, upper (75) and lower (25) quartiles. **J** Same as (**L**), but the Gal-ChR2 spike and Gal-EMG fidelity (with which a laser pulse triggered a single spike or EMG response) is plotted. Elements of violin plot: center line, median; box limits, upper (75) and lower (25) quartiles. **K** Violin plots illustrating the amplitudes of optogenetically time-locked (5 Hz) BSM signals (*N* = 12 mice) during 1st (mean amplitude 7.45 ± 1.64 mV), 2nd (mean amplitude 3.59 ± 1.17), 3rd (mean amplitude 2.93 ± 1.26 mV) and last rounds (mean amplitude 3.98 ± 1.98) of laser application, with boxplots (elements of violin plot: center line, median; box limits, upper (75) and lower (25) quartiles). *P*-values result from a two-tailed Mann-Whitney-U Test. Every dot is an individual animal. 11 out of 12 animals showed BSM responses upon 2nd laser application, in 9 out of 12 animals BSM responses were triggered upon 3rd laser application and in 8 out of 12 animals, laser application led to BSM responses during last rounds of stimulation (5th laser applications). **L** Same as (**I**), but the duration of BSM signals is plotted. Mean durations: after 1st 0.06 ± 0.01 s; 2nd 0.05 ± 0.01; 3rd 0.03 ± 0.01; and last 0.03 ± 0.06 laser application. *P*-values result from a two-tailed Mann-Whitney-U Test. Elements of violin plot: center line, median; box limits, upper (75) and lower (25) quartiles. **M** Same as (**I**), but the onset with which locked laser light triggered a BSM response. Data refers to 5 averaged onsets. Mean onsets: after 1st 0.03 ± 0.005 s; 2nd 0.03 ± 0.006; 3rd 0.03 ± 0.006; and last 0.03 ± 0.005 s, laser application. Elements of violin plot: center line, median; box limits, upper (75) and lower (25) quartiles. **N** Optogenetic activation of photo-identified BSM-MNs (*N* = 7 cells from 6 mice; same experimental approach as in Fig. 1G). Upper left panel: Juxtacellular recording of a BSM-MNs in parallel with EMG recordings of the BSM (2nd trace) during a 20 Hz laser stimulation. Upper right panel: zoom in of the indicated box illustrating the BSM-MNs spike and the EMG onset. **O** Repeated optogenetic activation of BSM-MNs (*N* = 10 mice) led to BSM activity of stable amplitude (mean amplitude: after 1st 7.65 ± 0.91 mV; 2nd 8.66 ± 0.93 mV; and 3rd 6.99 ± 0.59 mV laser applications). Elements of violin plot: center line, median; box limits, upper (75) and lower (25) quartiles. **P** Same as (**O**), but the duration of BSM activity in response to optogenetic activation of BSM-MNs (*N* = 10 mice) is shown (mean durations: after 1st 15.28 ± 0.47 ms; 2nd 15.85 ± 0.39 ms; and 3rd 15.61 ± 0.47 ms laser application). Elements of violin plot: center line, median; box limits, upper (75) and lower (25) quartiles.

Taken together, these experiments suggest that electrical stimulation of the mouse spinal cord in the area where the Gal+ neurons are located can only evoke the second and final phase of the ejaculatory process (expulsion), but not the emission phase. The differing outcomes of the electrical stimulations based on the male's internal state (Sexually Naive vs. 5 Mounts vs. Ejaculation) imply that the properties of neuronal circuits in this spinal cord region are modulated by copulation and ejaculation, suggesting that the spinal circuitry may be involved in controlling the refractory period in addition to ejaculation.

**Optogenetic stimulation of Gal+ neurons leads to BSM activity**
To address the inherent lack of specificity in electric stimulation experiments and to unequivocally link the activity of Gal+ cells, BSM-MNs and the BSM, we genetically restricted the neuronal population being stimulated to the Gal+ neurons by crossing the Gal-cre mouse line with the ChR2 line (Gal-ChR2[50]). To activate the Gal+ neurons, we placed an optical fiber on top of the spinal cord of anesthetized spinalized mice and delivered brief pulses of blue light along its rostrocaudal axis (Fig. 5A, left panel, *N* = 10 mice). Light delivery (20 mW,

200 Hz, 100 pulses with 10 ms duration) led to comparable responses to the ones elicited via electrical stimulation (Fig. 5B lower right panel). As expected, BSM activity was only observed in response to illumination at the location with the highest ChR2 expression (revealed by either DiI injections or an electrolytic lesion, Fig. 5B, see also Supplementary Fig. 14), specifically above the L2/L3 spinal segments (Fig. 5C, D), the region previously shown to harbor the highest density of Gal+ cells (Fig. 2E). Consistent with the results obtained from electric stimulations, optogenetically-evoked BSM responses were markedly higher in amplitude and longer in duration in spinalized compared to intact preparations (Supplementary Fig. 15), further supporting the idea that the Gal+ cells may be subject to tonic descending inhibition[14,16–18].

Furthermore, optogenetically elicited BSM responses exhibited similar dynamics to those evoked by electrical stimulation. Both methods resulted in a decrease in amplitude and duration with consecutive trains of stimulation (see example traces in Fig. 5B and population data in Fig. 5E, F), while the onset of triggered responses remained unchanged (Fig. 5G).

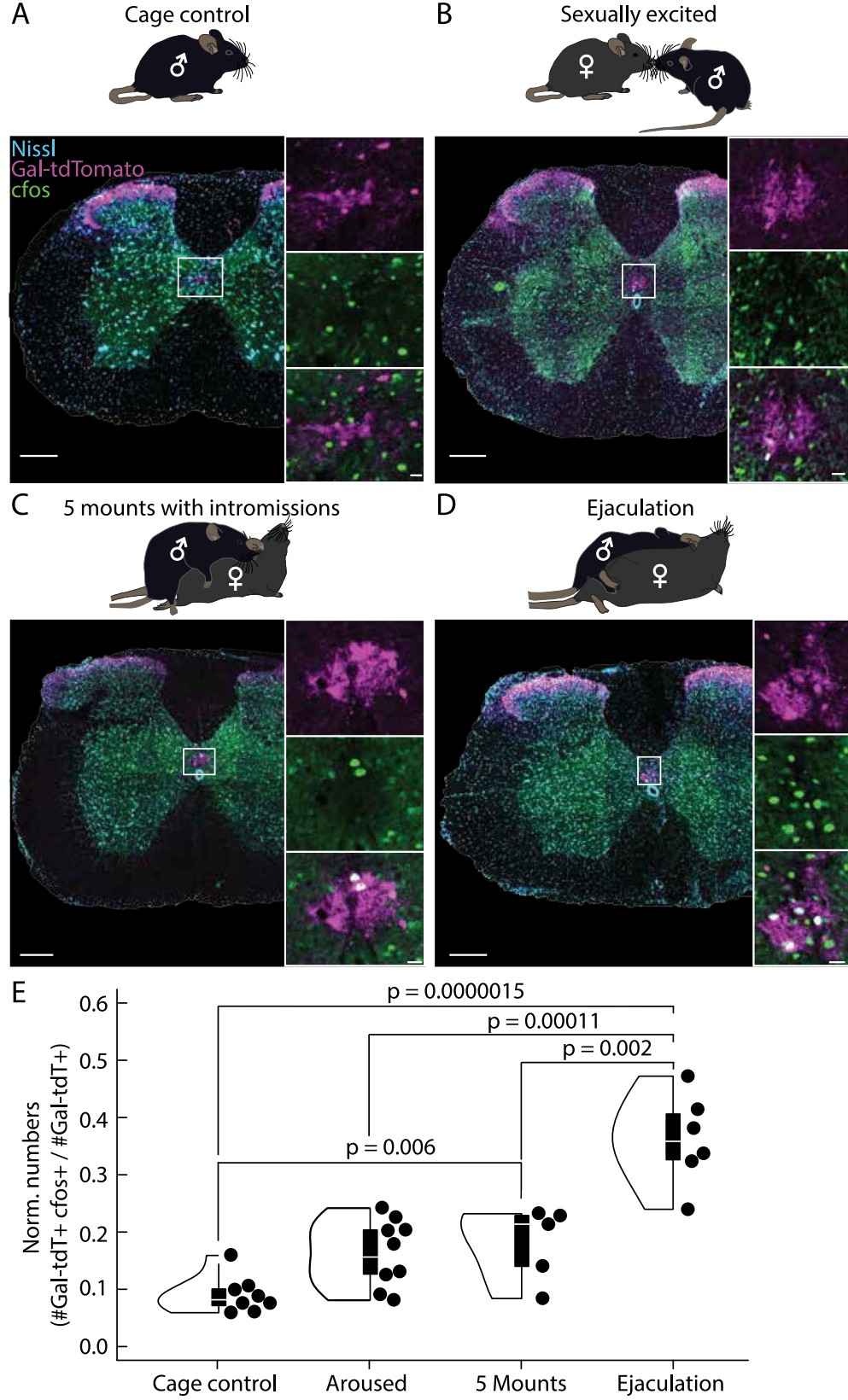

In addition to these bursts of activity, we used lower stimulation frequencies to trigger time locked BSM responses ($N = 12$ mice; 20 mW, 5–50 Hz, 15 pulses; Fig. 5H–M). In a subset of animals, we successfully conducted juxtacellular recordings in photo-identified single Gal-ChR2 neurons (Fig. H–M, $N = 7$ cells), revealing response dynamics upon optogenetic stimulation in individual Gal+ cells and the consequent dynamics of the BSM with time locked triggered activity. The mean latency to spike for Gal+ cells was 10.6 ms ± 1.7 ms, followed by light triggered BSM EMG with a mean latency of 6.7 ms ± 0.09 ms. Notably, the mean latency to spike for photo-identified BSM-MNs was 4.5 ms ± 0.4 ms; Fig. 5I), further supporting the existence of a monosynaptic connectivity between Gal+ cells and BSM-MNs, as demonstrated above

**Fig. 6 | The lumbar population of Gal+ neurons becomes increasingly active during sexual behavior. A** Upper panel: Males in the Cage control group were alone in their home cage or in the behavior box for 10 min ($N = 8$ mice). Lower panel: Example spinal cord section at the L2/L3 segments (revealed by Nissl stain, blue). Note no overlap between the cFos signal (green) and the Galanin (Gal) signal (purple; see larger insets). Scale bar spinal cord section 200 μm, scale bar inset 20 μm. **B** Upper panel: Males in the Excited group were allowed to interact with a hormonally primed ovariectomized female for 10 min, but attempts of copulation were interrupted ($N = 9$ mice). Lower panel: Example spinal cord section at the L2/L3 segments. Scale bar spinal cord section 200 μm, scale bar inset 20 μm. **C** Upper panel: Males in the 5 Mounts group: were allowed to perform 5 mounts with vaginal thrusting ($N = 5$ mice). Lower panel: Example spinal cord section at the L2/L3 segments. Scale bar spinal cord section 200 μm, scale bar inset 20 μm. **D** Upper panel: Males in the Ejaculation group were allowed to engage in sexual behavior until they ejaculated ($N = 6$ mice). Lower panel: Example spinal cord section at the L2/L3 segments. Scale bar spinal cord section 200 μm, scale bar inset 20 μm. **E** Normalized cell numbers (number of double labeled cells, cfos+ and Gal+; divided by the total number of Gal+ cells) for the four groups. Depicted $p$ values from two-tailed Student's t-test (elements of violin plot: center line, median; box limits, upper (75) and lower (25) quartiles).

through in vitro recordings (Fig. 2I–N). Spike and EMG fidelities (calculated as the number of spikes or EMG responses divided by the number of light pulses) were stable up to 10 Hz (Fig. 5J). Importantly and similar to the tonic discharges observed above, a decrease in amplitude (Fig. 5K) and duration (Fig. 5L) was observed with consecutive lower frequency optogenetic stimulation, with a constant onset of triggered responses (Fig. 5M).

Finally, the decrease in BSM activity observed with the electric and optogenetic stimulation of Gal+ cells contrasts sharply with the responses evoked by direct optogenetic activation of BSM-MNs. Direct optogenetic stimulation of BSM-MNs resulted in consistent responses in single BSM-MNs and in the BSM across repeated rounds of stimulation (Figs. 1G–N and 5N–P). This suggests that the depression in BSM activity upon consecutive stimulation of Gal+ neurons may be due to an alteration in synaptic transmission between the Gal+ cells and BSM-MNs, rather than changes in the intrinsic properties of BSM-MNs.

**Genetic ablation of Gal+ neurons disrupts copulatory behavior**

Given the unexpected finding that the outcome of electrical stimulation of the spinal cord on BSM activity was dependent on the behavioral state of the male, suggesting the involvement of Gal+ cells during other phases of copulatory behavior (see Fig. 4H–M), we used the expression of the immediate early gene *cFos* to establish a link between behavior and neuronal activity[59]. For this purpose, we compared the induction of the cFos protein in the Gal+ cells of the L2/L3 spinal segments of male mice that ejaculated (sexual interaction with a receptive female until ejaculation, Ejaculation, $N = 6$ mice) with the induction in males that were either alone in a clean cage (Cage control, $N = 7$ mice) or had varying degrees of interaction with a female (10 min with a receptive female but no penile insertion as intromission was interrupted whenever the male attempted copulation, Sexually Excited, $N = 5$ mice; or five mounts with intromission and intravaginal thrusting, 5 Mounts, $N = 5$ mice) (Fig. 6A–D).

To specifically quantify the number of active Gal+ cells, we used male progeny from the Gal-cre x tdTomato cross and performed post-hoc immunohistochemical quantification of the tdTomato signal and cFos induction in the L2/L3 spinal segments for each condition. As expected, given the direct contact to the BSM-MNs, significantly more double-positive neurons were found in the Ejaculation group, compared to the other three conditions (Cage control, $p = 0.0000015$, Sexually Aroused, $p = 0.00011$ and 5 Mounts, $p = 0.002$ Two-tailed Student's t-test, Fig. 6E). Unexpectedly, we observed similar levels of activation between the Sexually Excited and 5 Mounts groups, with both conditions exhibiting significantly higher levels of double-positive neurons compared to the Cage control ($p = 0.0067$, Two-tailed Student's t-test). To test if the activation observed in the Sexual Excited group was specific to an interaction with a female, we conducted a separate experiment evaluating cFos activation levels after introducing a male intruder in the cage with the test animal and compared it to the Cage control and Sexually Excited groups. The interaction with the intruder male, which involved bouts of aggressive behavior and male-directed mounting, elicited similar levels of cFos activation in the Male and Sexually Excited groups (Supplementary Fig. 16). Given the high arousal state of the male in these two

conditions, this result suggests that the activity of the Gal+ cells may be related to the internal state of the male and not specific for a sexual interaction.

Our results so far suggest that the population of lumbar Gal+ cells might have a role before the ejaculatory threshold. To determine the function of this spinal population during a full sexual interaction, we used a genetic approach to specifically ablate the Gal+ cells by expressing the diphtheria toxin receptor (DTR[60]). A conditional AAV carrying the DTR construct (AA8V-FLEX-DTR-GFP, Salk) was injected into the L2/L3 spinal segments of sexually trained males derived from the Gal-Cre x tdTomato cross (Fig. 7A; DTR group; $N = 12$ mice). The control group, of the same genotype and also sexually trained, underwent a sham surgery at the same spinal location (SHAM group; $N = 7$ mice).

After a recovery period and a second session of sexual behavior, both groups (DTR and SHAM) received an intraperitoneal injection of diphtheria toxin (DT). One week after DT treatment, the effect of the genetic ablation on sexual behavior was tested in the presence of a sexually receptive female. Males of both groups were sacrificed either 90 min after ejaculation or 90 min after the female was introduced into the testing arena if the male initiated copulation but did not ejaculate. In cases with absent sexual motivation (no mount attempts), the trial was interrupted 30 min after the female was introduced, and testing was repeated up to two more times, once a week (Supplementary Fig. 17A and see "Methods" for detailed experimental procedure[4]).

Post-hoc immunohistochemical processing of the spinal cords (Fig. 7B, C) allowed us to determine the number of cFos-positive and Gal+ cells in the L2/L3 spinal segments. While the number of cFos-positive cells was comparable between the SHAM and DTR group (Fig. 7B–D), there was a significant reduction in the number of Gal+ cells (Fig. 7E, G) and in the overlap of Gal+ neurons and cFos-positive cells (Fig. 7F, G). Consistent with the previous cFos and electrical stimulation experiments (Figs. 3 and 5) and further supporting the involvement of these neurons in a more general control of sexual behavior, the copulatory sequence (Fig. 7H) was significantly disrupted in the DTR group. While 3 out of 12 DTR animals did not reach ejaculation (but attempted copulation), and only one ejaculated in less than 10 min, all SHAM animals reached ejaculation, with only two taking longer than 10 min (Fig. 7I), as indicated in the cumulative distribution of the latency to ejaculate from the first mount (Mount with probing—MP or Mount with Intromission—MI). Individual raster plots for each animal aligned to the first consummatory act (Fig. 7J) illustrate no effect on the latency to mount (Fig. 7K), while the latency to ejaculate from first mount was significantly longer (Fig. 7L).

To identify which aspect of the sexual interaction was disrupted, leading to an increased latency to ejaculate, we further analyzed the copulatory sequence (including only DTR animals that ejaculated in the analysis). Sexual motivation did not differ across groups as reflected by the latency to mount (Fig. 7K) and the number of anogenital investigations, though there was a trend towards a higher number of anogenital investigations in the DTR group (Supplementary Fig. 17B), in particular when considering the number of events during the consummatory phase of the behavior (after the first mount, Supplementary Fig. 17B).

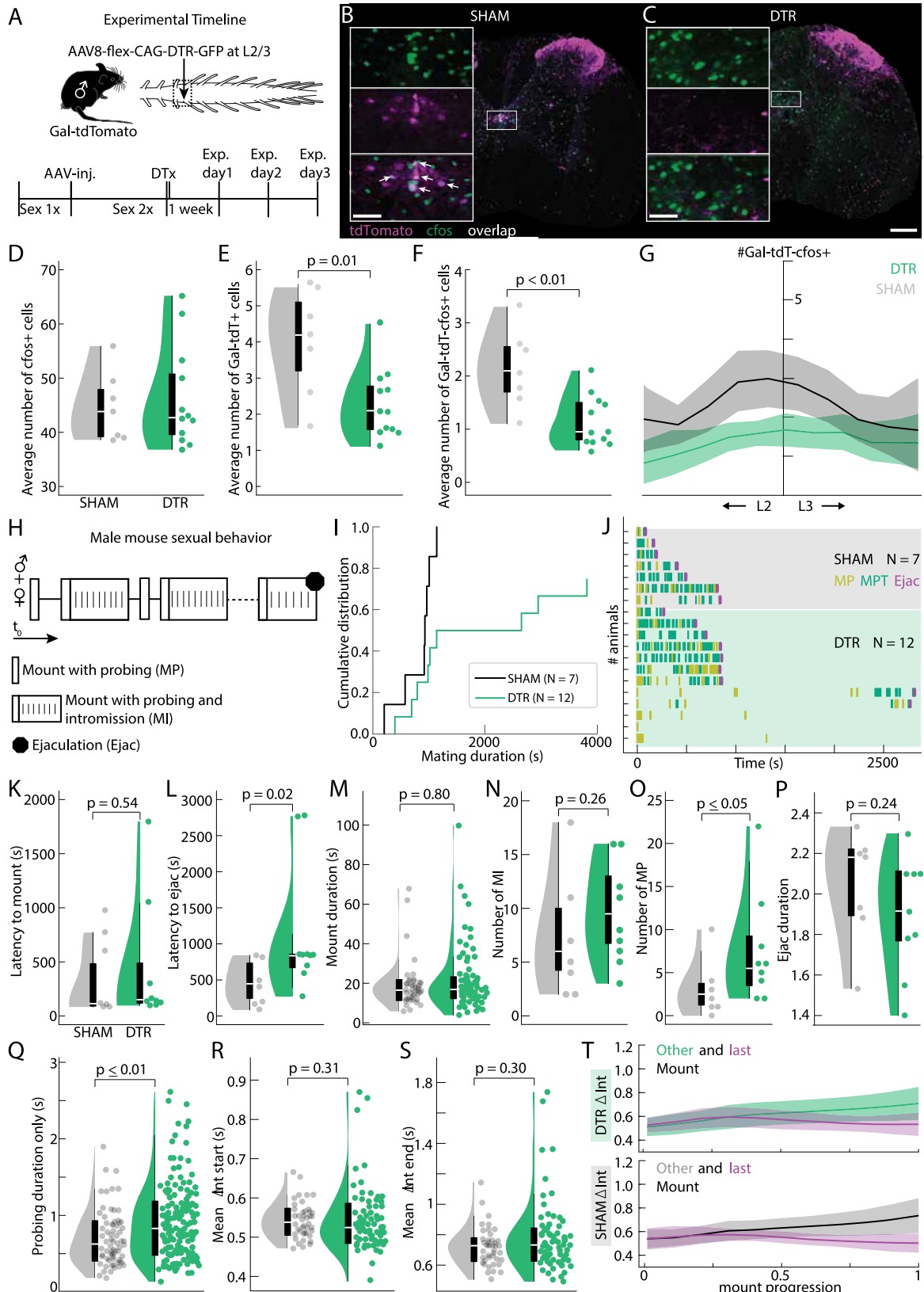

Interestingly the mounting events resulting in penile insertion (MI, Fig. 7H), had similar durations across groups (Fig. 7M and Supplementary Fig. 17C), and the number of MI was also similar (Fig. 7N), indicating that a comparable amount of genital sensory information was sufficient to trigger ejaculation in both types of males, despite differences in the latency to ejaculate. In contrast, we observed a difference in the number of mounts that did not result in penile insertion: mount attempts where the male performed several shallow pelvic thrust movements trying to locate the female's vagina (which only occur if the male is sexually motivated, MP; Fig. 7O[61]). Surprisingly, the ejaculation duration was similar across all males that ejaculated (from the last thrust

**Fig. 7 | Genetic ablation of the lumbar Gal+ neurons disrupts male sexual behavior. A** Experimental design. Gal-cre x dTomato male mice were sexually trained once before receiving a spinal injection of either a flexed AAV carrying the diphtheria toxin receptor (DTR) (DTR group) or undergoing a sham surgery (SHAM). Two weeks post-surgery the sexual performance of both groups was tested, after which they received an injection of Diphtheria Toxin (DT). One week after DT injection, the impact of the neuronal ablation on sexual behavior was investigated. **B** Example spinal cord section at the L2/L3 segments obtained from an animal of the SHAM group. tdTomato indicating Galanin-positive (Gal+) neurons (purple) overlap (white) with the immediate early gene cFos (green). Scale bar 200 μm, inset 50 μm. **C** Same as (**B**) but for an animal of the DTR group. Note the poor TdTomato signal (purple) around the central canal at the L2/L3 spinal segments. **D** Violin plots of cFos cells quantified in the L2/L3 spinal segments for the SHAM (gray, mean number 44.6 ± 2.4, $N = 7$ mice) and DTR (green, mean number 46.2 ± 2.1, $N = 12$ mice) animals. $P = 0.7$ resulting from a two-tailed student's t-test (elements of violin plot: center line, median; box limits, upper (75) and lower (25) quartiles). **E** Same as (**D**), but the average number of Gal-tdT+ cells is depicted. Note the significantly lower number of Gal-tdT+ cells in the DTR animals (mean number of tdTomato+ cells 2.3 ± 0.28, $N = 12$ mice) compared to the SHAM group (4.0 ± 0.6, $N = 7$ mice). $P = 0.01$ resulting from a two-tailed student's t-test. Elements of violin plot: center line, median; box limits, upper (75) and lower (25) quartiles. **F** Same as (**D**), but the average number of Gal-tdT+ cells co-expressing cfos is plotted. A significant difference was observed across groups (mean number of Gal-tdT+/cfos+ cells in DTR 1.1 ± 0.14 ($N = 12$ mice) and SHAM 2.1 ± 0.3 animals ($N = 7$ mice), $p = 0.01$ resulting from a two-tailed student's t-test). Elements of violin plot: center line, median; box limits, upper (75) and lower (25) quartiles. **G** Distribution of Gal-tdT+ cFos+ cells along the L2/L3 spinal segments for SHAM ($N = 7$ mice) and DTR animals ($N = 12$ mice); solid line, bootstrapped median for all the animals; shaded error bars, bootstrapped interquartile range. **H** Schematic representation of male sexual behavior: Mounts with probing (MP), correct positioning of the paws on the female flanks and shallow pelvic thrusting movements trying to locate the vagina; Mounts with Intromission (MI), after the initial probing period of shallow thrusting the male inserts the penis inside the female and executes several deeper thrusts until he dismounts; several MI are executed until the ejaculatory threshold is achieved (Ejac). **I** Cumulative distribution of the latency to ejaculate from the first consummatory act (MP or MI) (black SHAM ($N = 7$ mice); green DTR ($N = 12$ mice). Proportion of animals that ejaculated (y-axis) within a given time (x-axis) is shown. Survival analysis using the two-tailed Kaplan–Meier estimator shows a significant difference between the SHAM and DTR distributions by the log-rank test ($p = 0.02$). **J** Raster plot aligned to the first consummatory act (MP or MI). Each line represents an animal: SHAM animals on a shaded gray ($N = 7$ mice); DTR animals on a shaded green background ($N = 12$ mice). Purple line marks Ejaculation (Ejac). Animals are

ordered by latency to ejaculate, after separating animals that ejaculated or not. **K** Violin plot showing the latency to mount (MP or MI) for SHAM (mean latency 392 ± 181 s, $N = 7$ mice) and DTR (mean latency 435 ± 248 s, $N = 9$ mice) animals. No significant difference was detected, $p = 0.54$, two-tailed Mann-Whitney U-test (elements of violin plot: center line, median; box limits, upper (75) and lower (25) quartiles. **L** Same as (**K**), but the latency to ejaculate from the first consummatory act (MP or MI) is plotted (mean latency to ejac for SHAM animals 427.2 ± 146 s, $N = 7$ mice). Note that DTR animals take significantly longer to reach ejaculation (mean latency to ejac 1170.8 ± 392 s, $N = 9$ mice). $P = 0.002$ resulting from a two-tailed Mann-Whitney U-test. Elements of violin plot: center line, median; box limits, upper (75) and lower (25) quartiles. **M** Same as (**K**), but for the mount duration (all MI events) which is comparable across groups (mean mount duration of SHAM animals 18.9 ± 2 s ($N = 7$ mice) and DTR animals 21.1 ± 2 s ($N = 9$ mice)). $P = 0.8$ resulting from a two-tailed Mann-Whitney U-test. Elements of violin plot: center line, median; box limits, upper (75) and lower (25) quartiles. **N** Number of MTs for SHAM (mean number 7 ± 2.7, $N = 7$ mice) and DTR (mean number 9.3 ± 2, $N = 9$ mice) animals. $P = 0.29$ resulting from a two-tailed Mann-Whitney U-test (elements of violin plot: center line, median; box limits, upper (75) and lower (25) quartiles. **O** Number of MPs for SHAM (mean number 3 ± 1.6, $N = 7$ mice) and DTR (mean number 7.6 ± 2.7, $N = 9$ mice) animals. $P = 0.03$ resulting from a two-tailed Mann-Whitney U-test (elements of violin plot: center line, median; box limits, upper (75) and lower (25) quartiles. **P** The time window for ejaculation is plotted (from the last thrust until separating from the female). Mean ejac duration for SHAM animals 16 ± 3 s ($N = 7$ mice) vs. mean DTR 15.4 ± 2.3 s ($N = 9$ mice). $P = 0.8$ resulting from a two-tailed student's t-test (elements of violin plot: center line, median; box limits, upper (75) and lower (25) quartiles. **Q** Probing duration for all MI (time from first to the last shallow thrust, before penile insertion). Mean probing duration for SHAM 0.7 ± 0.04 s ($N = 7$ mice) vs. DTR 0.91 ± 0.04 s ($N = 9$ mice). $P = 0.002$ resulting from a two-tailed Mann-Whitney U-test (elements of violin plot: center line, median; box limits, upper (75) and lower (25) quartiles. **R** Violin plots of all inter-thrust intervals (ΔInt) of the first four thrusts of a MI (mean for SHAM 0.54 ± 0.01 s ($N = 7$ mice) vs. DTR 0.54 ± 0.01 s ($N = 9$ mice). $P = 0.31$ resulting from a two-tailed Mann-Whitney U-test (elements of violin plot: center line, median; box limits, upper (75) and lower (25) quartiles. **S** Same as (**R**), but the inter-thrust interval of the last four thrusts in a MI is plotted (mean for SHAM 0.69 ± 0.03 s ($N = 7$ mice) vs. DTR 0.76 ± 0.03 ($N = 9$ mice). $P = 0.15$ resulting from a two-tailed Mann-Whitney U-test. Elements of violin plot: center line, median; box limits, upper (75) and lower (25) quartiles. **T** Mean interval with standard deviation is plotted in relation to mount progression for all other MI (light green DTR ($N = 7$ mice); gray SHAM ($N = 9$ mice)) and the last MI leading to ejaculation (purple); a similar pattern of deceleration and acceleration is observed across the two groups.

until the male dismounted the female, see "Methods"; Fig. 7P), indicating that if the male reached the ejaculatory threshold, sperm expulsion could occur. This contrasts with results from similar experiments in rats, where ablation of rat Gal+ cells disrupted ejaculation, but not copulation[21]. In line with the increased number of MP, the time to successful penile insertion after the male placed his paws on the female flanks was also significantly longer in the DTR group (Fig. 7Q, Supplementary Fig. 17D). Thrusting rate and dynamics of MI were similar across groups (Fig. 7R–T), indicating normal pelvic thrusting once penile insertion was achieved. Importantly, erection did not seem affected by the ablation of Gal+ cells, as penile movements were similar in SHAM and DTR animals, with both groups capable of penile insertion and intravaginal thrusts (as observed in the Supplementary Movies S3 and S4).

Finally, to test if the behavioral impairments were due to the extent of the ablation, we assessed the relationship between the number of double-positive neurons (Gal+ and cFos+) and various behavioral parameters. There was no correlation between the number of MI, total number of thrusts, mount duration (Supplementary Fig. 18A, B), or other behavioral parameters (Supplementary Fig. 18C–J) and the double-positive neurons, meaning that the magnitude of behavioral impairments is not only explained by the extent of ablation, but perhaps by other compensatory mechanisms discussed below.

Taken together, our results indicate that the ablation of the lumbar Gal+ population leads to an increase in the latency to ejaculate and a significant disruption of the copulatory sequence, supporting a more complex involvement of the spinal cord in controlling sexual behavior in mice.

## Discussion

In mammals, pre-copulatory and copulatory actions orchestrated by the brain are thought to bring the male to the ejaculatory threshold, such that penile insertion in the female's vagina triggers ejaculation and sperm ejection, a reflex controlled by a spinal cord circuitry. This assumption implies that the brain plays no role in the ejaculatory reflex (other than inhibiting it until the threshold is reached) and that the activity of the spinal network is inconsequential to the orchestration of copulatory behavior (serving only as a relay of ascending sensory input). However, this division of labor between the brain and the spinal cord has been repeatedly challenged, particularly by the existence of neurological disorders suggesting that spinal neurons are integral to the control system, with central and effector players in continuous information exchange[62,63]. Our study provides evidence supporting the involvement of the spinal cord in controlling male mice's internal state and copulation, rather than merely relaying penile stimulation and controlling ejaculation. Using the penile muscle involved in sperm expulsion as a point of entry, we have characterized a microcircuit in

the lumbar spinal cord of male mice consisting of motor neurons innervating the bulbospongiosus muscle (BSM-MNs) and a population of galanin-positive (Gal+) neurons that is monosynaptically connected to the BSM-MNs. Moreover, we provide strong anatomical evidence for a reciprocal connection between the Gal+ neurons and autonomic nuclei, as well as the ischiocavernosus MNs. This suggests a potential role of the Gal+ population not only in emission but also in erection[45,64–66]. The output to the ischiocavernosus MNs might be important for the regulation of penile detumescence, which must be finely regulated at the time of ejaculation, as pressure has to momentarily decrease for expulsion to take place[67].

The population described in our study shares a similar molecular profile with the previously identified rat Spinal Ejaculator Generator (SEG[37–39]), despite some differences in the spinal segments where the Gal+ neurons are located (L2/L3 segments versus L3/L4 in the rat[21]). Several lines of evidence supported labeling the rat Gal+ population as the SEG. While electrical stimulation at the location of the putative SEG evokes ejaculation in anesthetized rats[56], its ablation results in complete disruption of the ejaculatory reflex without affecting copulatory behavior[21]. Furthermore, the SEG seems to be anatomically connected to the sensory branch of the pudendal nerve, indicating it received genital information[53,54]. However, to the best of our knowledge, the functional connectivity between the rat SEG and the BSM-MNs has never been established.

Our study, using a dual and genetic-based approach combined with in vitro whole-cell patch-clamp recordings of BSM-MNs while specifically activating Gal+ axons, provides data unequivocally establishing a functional monosynaptic connection between Gal+ cells and BSM-MNs. Despite the molecular similarities between the Gal+ neurons described in this study and the rat SEG, our results indicate that the properties of the mouse spinal network may differ significantly from its rat counterpart.

First, electrically induced ejaculation in anesthetized rats produces similar motor and physiological activity patterns regardless of an intact connection to the brain (which we reproduced in our study), while penile stimulation can activate ejaculation only in anesthetized spinalized rats[15]. These findings suggest the presence of descending inhibition that gates sensory input from the penis and that the rat SEG can function independently of the brain once this inhibition is removed. In contrast, in mice, pronounced BSM activity was only triggered when the Gal+ cells were electrically or optogenetically activated in a spinalized preparation, but not if the connection to the brain was intact. Moreover, penile stimulation did not trigger an ejaculation in our experiments. Therefore, our results support a model where, at least in mice, the Gal+ population and the incoming sensory input is kept under a substantial inhibition from the brain, likely from the ipsilateral paragigantocellular thalamic nucleus[20], until the ejaculatory threshold is reached.

We were intrigued by our inability to trigger emission in mice through Gal+ and penile stimulation, despite demonstrating an anatomical connection to autonomic centers, the main players in the emission process[68], in contrast to what was observed in the rat. While we cannot rule out the possibility that our failure to trigger emission was due to suboptimal stimulation parameters, we propose that these results are biologically plausible and may reflect species differences in the regulation of ejaculation.

Even though we could not induce emission through artificial stimulation of the spinal cord location containing the Gal+ cells or by directly activating the Gal+ neurons, our EMG recordings in behaving animals showed that artificially-induced BSM activities closely resembled natural BSM activity, further supporting that our stimulation parameters elicited activity within physiological ranges and that Gal+ stimulation alone is insufficient to induce emission.

Mouse copulation is characterized by repeated vaginal thrusting leading to a single ejaculation, a copulatory sequence more similar to humans but quite different from rats, which have multiple ejaculations dependent on the execution of multiple individual penile insertions. This suggests that the buildup to the ejaculatory threshold in rats may differ from mice and humans, as it might be independent or much less influenced by the integration of genital sensory input, while in the mouse, sensory integration of multiple thrusts is crucial. While rat data suggest that descending inhibition operates primarily at the level of the SEG, our data points for the descending inhibition to modulate the Gal+ neurons and the incoming sensory input to them in mice.

While anatomically the circuitry seems to be similar between rats, mice and humans, the functional logic may be fundamentally different. The rat spinal circuitry appears to function as a true spinal reflex arc, capable of driving ejaculation with a sensory stimulus if disconnected from the brain. In contrast, the mouse circuit does not qualify as a proper reflex since the sensory input alone cannot drive emission or full BSM activity, supporting a constant dialogue between the brain and the spinal cord, for the buildup of sexual excitation and ejaculation. Future comparative experimental efforts are needed to reveal the operational logic of these circuits in different species. For example, it remains to be determined if a true central pattern generator for ejaculation exists in the mouse and how the Gal+ neurons we have identified connect to such network. However, the discrepancies outlined in this study may simply result from different modus operandi, the product of different evolutionary trajectories. We observed an unexpected gradual increase in the activity of the Gal+ population during sex and when interacting with another male, suggesting the activity of these neurons is related to the arousal level of the individual. This contrasts with previous studies in rats, which showed that Gal+ cells are only active after an ejaculation[69]. To our surprise, repeated electrical stimulation was accompanied by a marked decrease in BSM activity, which was not due to a deterioration of the preparation. Moreover, if male mice were allowed to have sex and ejaculate just before the electric stimulation experiment, BSM activity was significantly lower and resembled the activity observed after multiple rounds of stimulation. These results indicate the Gal+ neurons are recruited not only during ejaculation but also in response to arousal and sexual activity. This supports the notion of a continuous dialogue between the spinal cord and the rest of the body and a possible prominent role of the spinal cord in the control of the refractory period[3], contrary to what is currently hypothesized.

Lastly, the chemogenetic ablation of the Gal+ population caused a disruption in sexual behavior. Interestingly, and in contrast to the rat SEG, whose ablation led to a complete abolishment of ejaculation while leaving copulatory patterns intact[21], only 3 out of 12 animals in our study did not ejaculate. However, since the session was artificially interrupted after a certain time limit, it is unknown if they would have eventually ejaculated given more time. Also, and contrary to the rat, we observed altered sexual behavior in all animals, including increased latency to ejaculate, more mounts with probing, and longer durations of probing, suggesting that immediate sensory penile feedback is partially disrupted when Gal+ cells are ablated.

These results raise two immediate questions: how can animals still ejaculate if the Gal+ neurons that contact the BSM-MNs are ablated, and how and where is the penile sensory feedback processed to reach ejaculation? The first question may be explained by a small percentage of Gal+ cells not being ablated, which could suffice to integrate pelvic sensory input and drive ejaculation. Additionally, redundant spinal circuits may compensate for the loss of Gal+ cells. Namely, circuits controlling pelvic floor muscles and organs, responsible for sexual behavior, micturition and defecation, share common connections between spinal and brain nuclei[70]. For instance, connections between the sacral parasympathetic nuclei that control pelvic organ function (i.e., emission), and the BSM-MN nucleus have been described in cats[71,72] and rats[41]. Holstege and Tan (1987) hypothesized that BSM-MNs are a special class of neurons with both somatic and autonomic functions, capable of controlling both processes[72].

Moreover, direct brain spinal projections, namely to the BSM-MNs, may drive sperm expulsion independently of Gal+ cells. Anatomical evidence indicates a direct connection between the hypothalamic paraventricular nucleus (PVN) and the BSM-MNs in cats[72] and rats[73,74]. This pathway appears to regulate erection but also rat sexual behavior in general[75]. Additionally, in rats, ejaculation can be facilitated by an oxytonergic PVN projection onto SEG cells[76]. Projections from the Barrington's nucleus, also known as the pontine micturition center, connect to pelvic-related nuclei in the spinal cord (in rats[77], cats[78] and humans[79]), and electrical stimulation of the medial-lateral areas induced emission and expulsion-like responses, respectively[78].

Regarding the sensory relay, studies in rats and humans showed that pressure increase in the prostatic urethra caused by the arrival of sperm and seminal fluid at the time of emission is sufficient to induce ejaculatory-like contractions in the BSM[80,81]. Rat dorsal penile nerve to BSM-MNs connectivity has been described[82–84] and might be sufficient to relay sensory information, without the Gal+ neurons in our experiments. Finally, Qi et al. (2023) described Krause corpuscles as relayers of genital sensory information in mice, and their manipulation impacted sexual behavior, reinforcing the idea that sensory feedback may be necessary to sustain copulation and ejaculation[85]. As for the second question, it remains unclear where arousal is controlled and how sensory input impinges on such circuits. But our results point towards an involvement of the spinal network. In summary, we have identified a cluster of Gal+ cells in the lumbar spinal cord of mice that are directly connected to BSM-MNs and that seem to be involved in the integration of sensory signals during copulation and the male's internal state, thereby taking a more central and intricate involvement in the control of mice copulatory behavior than a simple reflex arc.

## Methods

### Experimental model and subject details

All experimental procedures were carried out in strict accordance with the guidelines of the European Committee Council Directive and were approved by the Animal Care and Users Committee of the Champalimaud Neuroscience Program, the Portuguese National Authority for Animal Health (Direcção Geral de Veterinária; approval number 0421/000/000/2022) and by the local ethic committee of the University of Bordeaux and the French Agriculture and Forestry Ministry for handling animals (approval number 2016012716035720). For tracing from the BSM (FG and PRV), electrical stimulation experiments and EMG recordings in behaving animals, C57BL6 (*Mus musculus domesticus*, C57BL/6J) male mice aged 3–6 months were used. For tracing from the spinal cord, the cFos experiment and the DTR ablation, Gal-cre x tdTomato (*Mus musculus domesticus*, B6;129S6-Gt(ROSA)26Sortm9(-CAG-tdTomato)Hze/J) male mice aged 3–6 months old were obtained from Jackson Laboratories. For the spinal cord optogenetic stimulation and airpuff/electrical sensory stimulation of the penis, Gal- ChR2 (*Mus musculus domesticus*, B6;129S-Gt(ROSA)26Sortm32(CAG-COP4*H134R/EYFP)Hze/J) male mice aged 3–6 months old were obtained from Jackson Laboratories. For electric stimulation of the spinal cord, Long Evans (Crl:LE Long-Evans) male rats were ordered from Charles Rivers. For the pup muscle injections, C57BL6 male mice pups aged P3-P6 were used. For the pup optogenetic and electrophysiology experiment, Gal-ChR2 male mice pups aged P2-P6 were used. Finally, as a sexual stimulus, after ovariectomy and hormonal priming, C57BL6 female mice aged 3–6 months were used. All the animals were bred and maintained in our animal facility. Except for the optogenetic experiment in pups, all animals were weaned at 21 days and housed in same-sex groups in stand-alone cages (1284L, Techniplast, 365 × 207 × 140 mm) with access to food and water ad libitum. All experimental mice were maintained in suites with controlled temperature and humidity (22 celsius, 50% humidity), and on an inverse 12:12 light/dark cycle. Experiments were performed during the dark phase of the cycle, phase of higher animal activity. After initiation of

sexual behavior training or immediately after surgery, male mice were kept single-housed until the experiment was over. All animals were randomly allocated into the different experimental groups tested in this study.

### Bulbospongiosus muscle (BSM) injections

Mice were anesthetized with 3% isoflurane in oxygen and put into the mouthpiece of a stereotaxic device (Kopf, Tujunga, CA, USA). After that, mice were turned into a supine position to facilitate access to the BSM. After shaving the anogenital area and cleaning with Betadine (MEDA-Pharma) and 70% ethanol, a small incision in the scrotum area was made. At this point, an analgesic (buprenorphine, 0.05–0.1 mg/kg, intraperitoneal injection) was administered. The BSM was exposed after removing fat and conjunctive tissue. Pulled capillaries (length 3 1/2 inches [9 cm]; inner diameter 0.53, outer diameter 1.14 mm; tip diameter 40 µm; DrummondScientific, Broomall, PA, USA) were used to inject 1.5 µl of 2% Fluorogold (FG) at a rate of 13.8 nL per pulse, and a frequency of 0.2 Hz. In total, 5 BSM sites were injected with FG (three in the dorsal and two in the ventral portion of the BSM). A waiting time of 5 min prior to and 10 min after injection was kept. The glass pipette was pulled out slowly and the skin sutured. All mice were single housed post-surgery, for 1 to 2 weeks, until perfusion. A total of 12 male mice was used for anatomical investigation of BSM motor neurons using FG.

Another 6 male mice were injected with PRV-Ka-gEI-mCherry (PRV[32]) in the BSM. The male was similarly anesthetized with 3% isoflurane in oxygen and put into the mouthpiece of a stereotaxic device. After that, mice were turned into a supine position to access the BSM, and after disinfecting, a small incision was made to expose the muscle. Using a glass pipette as described above, 1 µl of PRV was injected once into the BSM at a rate of 13.8 nL per pulse, and a frequency of 0.2 Hz. A waiting time of 5 min prior to and 10 min after injection was kept, after which the pipette was pulled and the incision sutured. All mice were single housed post-surgery, for 3–4 days, until perfusion.

### Spinal cord stereotaxic viral injections and histology

Mice were anesthetized with 3% isoflurane in oxygen and spinally fixed into a stereotaxic frame using adapted vertebrae clamps (Kopf, Tujunga, CA, USA). During surgery, anesthesia was maintained using 1.5% isoflurane. Injection sites (lumbar segments 2/3) were targeted by using vertebral landmarks as described in ref. 86. Muscle and fat tissue were gently removed to get a better sight onto the spinal cord. The injection pipette was inserted in between the thoracic vertebrae T11 and T12 before having made a small puncture into the dura mater that allowed for better insertion of the injection pipette. Pulled capillaries (length 31/2 inches [9 cm]; inner diameter 0.53, outer diameter 1.14 mm; tip diameter 40 µm; DrummondScientific, Broomall, PA, USA) were used to inject the AAVs at 100 µm medial from the midline, at a depth of 750–850 µm from the spinal cord surface at a rate of 0.1 Hz with 2.3–4.6 nL per pulse. For the AAV1-CAG-floxed-SynGFPrev-WPRE (*N* = 7 mice), 30 nL were injected, whereas for the pAAV8-FLEX-DTR-GFP (*N* = 12 mice), 150–300 nL were injected either in one or two locations of the spinal cord. Before and after the pressure injection a waiting time of 10 min was kept. Afterwards, the pipette was retracted, eventual bleeding stopped and the skin sutured. Furthermore, as a control for the DTR experiment, sham injections were performed (*N* = 7 mice). Using the same method described above, the animals were anesthetized and spinal clamped to a stereotaxic frame. After fat and tissue removal to expose the vertebrae, a glass pipette without virus was lowered to the spinal cord at the Gal+ cells location. The pipette was kept inside the spinal cord for a total of 25 min, after which it was carefully removed, and the animal was sutured. Analgesic (buprenorphine 0.1 mg/kg) was always administered post-surgery. After sufficient time for viral expression (2 weeks for the pAAV8-FLEX-DTR-GFP and 3 weeks for the AAV1-CAG-floxed-SynGFPrev-WPRE) and eventual behavioral experiments (DTR experiment), animals were

deeply anesthetized and perfused transcardially with saline, followed by a cold 4% paraformaldehyde solution (PFA) in 0.01 mol/L PBS. Spinal cords were removed from the spine and kept for 1 h in 4% PFA before transferring them for another hour into 0.01 M PBS. Subsequently, spinal cords were stored overnight in 30% sucrose in 0.01 M PBS, 0.1% azide to cryo-protect the tissue. Spinal cords were embedded in frozen section medium and frozen for half an hour at −80 °C in 2-methylbutane solution before mounting them in the cryostat. Spinal cord sections which did not undergo subsequent immunohistochemical staining were cut and mounted on a poly-lysine-coated glass slide at 50 μm, sections with post-hoc immunohistochemical staining were cut and mounted at 30 μm.

### Pup viral injections

To infect the BSM-MNs with the light activated channel, channelrhodopsin (ChR2), young animals needed to be used as it has been shown that viral tracers are not able to infect the motor end plate beyond a certain age[29]. Pups (P3-P6) were briefly separated from their litter and mother and placed in a freezer. After all reflexes were gone, pups were placed on ice and a small incision below the penis was made to access the BSM. Pulled capillaries (length 3 1/2 inches [9 cm]; inner diameter 0.53, outer diameter 1.14 mm; tip diameter 15−20 μm; DrummondScientific, Broomall, PA, USA) were used to inject 1 μl of retroAAV-CAG-hChR2-H134R-tdTomato (28017-AAVrg, Addgene) into the BSM at a rate of 18.4 nl/pulse per second.

After injection, the incision was glued and the pups placed onto a heating pad. Once all reflexes were recovered, the pups were put back to their litter and mother. Injected pups were raised until 2–3 months of age before performing the acute optogenetic experiments as described below.

### Electromyogram (EMG) electrodes implantation for chronic recordings

For chronic BSM EMG recording in behaving mice, the animals were implanted with Myomatrix electrodes (model LNT-4×8-BVS-14) kindly made available to us from Sam Sober's Lab[87]. Mice were anesthetized with 3% isoflurane in oxygen and mouth fixed into a stereotaxic frame (Kopf, Tujunga, CA, USA), allowing the rotation of the body in order to implant the arrays in the bulbospongiosus muscle. During surgery, anesthesia was maintained using 1.5% isoflurane. Three incisions were made to implant the array: first in the pelvic area, then above the animal's right hind leg and finally a midline incision in the scalp. The skin was carefully separated from the muscle using scissors, using the three incisions to insert the scissors, and finally allowing to subcutaneously guide the arrays's connector from the pelvic area to the skull to be later secured. Once each thread has been routed subcutaneously and positioned near the BSM, the animal is turned into a supine position and each thread is inserted into the muscle using a suture needle. A suture (size 8−0) was tied to a hole in the array and the needle was then passed through the BSM and used to pull the attached array thread into the muscle. The array was then secured within the muscle by suturing it to the BSM, using additional holes on the array. Afterwards, the incision in the pelvic area was closed and the animal was turned to access the skull. The skull was carefully cleaned and disinfected, and the array's connector was secured to it using dental cement. If necessary, the incision on the skull was sutured and, after guaranteeing that all the array was under the skin, the incision above the hind leg was also sutured. All mice were single housed post-surgery and used for behavior afterwards.

### Nissl stain

The slides were washed 2 times in 0.01 M PBS to remove the excess of frozen section medium. After that, the sections were rehydrated for 40 min in PBS 0.1 M, pH 7.2 (PBS 10x) and permeabilized for 10 min with 0.1% Triton-X (T9284-100ML, Sigma-Aldrich) in PBS 10x. The

tissue was washed 2 times 5 min with PBS 10x before incubating them for 20 min with a 1:100 Neurotrace staining solution (in PBS 10x; (NeuroTrace™ 500/525 Green Fluorescent Nissl Stain, N21480, ThermoFisher Scientific; or NeuroTrace™ 530/615 Red Fluorescent Nissl Stain, N21482, ThermoFisher Scientific; or NeuroTrace™ 640/660 Deep-Red Fluorescent Nissl Stain, ThermoFisher Scientific). Subsequently, the tissue was washed with 0.1% Triton-X in PBS 10x for 10 min. After washing 2 times 5 min with PBS 10x, the slides were rinsed with distilled water, dried and coverslipped with Mowiol.

### Immunohistochemistry

Immunohistochemical labeling was performed using standard procedures. Briefly, spinal cord sections, which were labeled either for Gastrin releasing peptide (Gastrin Releasing Peptide (GRP) (Porcine) - Antibody, Phoenix Pharmaceuticals, H-027-13, Lot# 01742-1), Substance P (Anti-Substance P Receptor Antibody, Sigma-Aldrich, AB15810, Lot# 3022869), Enkephalin (Anti-Enkephalin/ENK antibody, Abcam, ab85798), CCK (Polyclonal Rabbit anti-Human CCK / Cholecystokinin Antibody, LSBio, LS-C190673, aa26-33), Osteopontin (Mouse Osteopontin/OPN Antibody, R&D Systems, AF808, Lot# BDO0617401), VGLUT1 (VGLUT 1 antibody, Synaptic Systems, 135 304, Q62362) or Galanin (Anti-Galanin Antibody, Milipore, AB2233, Lot# 3096488), were firstly washed 2 times 5 min with PBS 0.01 M to remove the excess OCT. Afterwards, they were washed 2 times 10 min with PBS 10x and preincubated for 1.5 h at room temperature in a blocking solution (PBS 10x, 1% bovine serum albumin, and 0.3% Triton X-100). Afterwards, primary antibodies were diluted in the same blocking solution at a proportion of 1:100. The primary antibody was incubated on the glass slides overnight at room temperature. Incubation with the primary antibody was followed by 5 times 10 min washing with PBS 10x. Subsequently, we proceeded to detect the primary antibody with a secondary antibody coupled to different fluorophores (Alexa Fluor 488, 594 or 647, Abcam/Thermo Fisher Scientific). The secondary antibody was diluted (1:500) in blocking solution and the reaction was allowed to proceed for 2 h in the dark at room temperature. In some cases, a Nissl stain was performed as described above. After the staining procedure, sections were washed 5 times 10 min with PBS 10x, rinsed with distilled water, dried and cover slipped with Mowiol mounting medium. Immunohistochemical staining for the immediate early gene cFos deferred slightly from the above-described procedure. Namely, after washing with PBS 0.01 M and PBS 10x, the sections were incubated in a different blocking solution (PBS 10x, 0.3% Triton X-100, 4% normal donkey serum, 1% bovine serum albumin) for 1 h at room temperature. The primary antibody (rabbit anti-cFos, Synaptic Systems, 226 003, Rb108B5), diluted 1:500 in blocking solution was added for 2 overnights at 4 °C. Afterwards, the sections were washed 3 times 5 min in PBS 10x, 0.3% Triton X-100, and the secondary antibody (Alexa Fluor 488 or 647, ThermoFisher Scientific) was added in blocking solution, diluted 1:500, for 2 h at room temperature. Finally, the sections were washed 3 times 5 min in PBS 10x, 0.3% Triton X-100, rinsed with distilled water, dried and cover slipped with Mowiol mounting medium. As described before, some slides were also counterstained with Nissl.

### Electrophysiology

**In vivo electrical stimulations and optogenetic stimulations.** For acute electrophysiological experiments, mice ($N = 8$) and rats ($N = 4$) were anesthetized by injection of an initial dose of 100 mg/kg ketamine and 7.5 mg/kg xylazine. Respiration, blink and pinch reflex were observed throughout the experiment and, if needed, animals were injected with an extra shot (25%) of ketamine/xylazine mixture or a 25% dose of ketamine alone. The animal's back, scrotum and right leg were shaved and cleaned with ethanol. Electromyogram electrodes were inserted into the BSM or tibialis anterior (TA) muscles and glued using Vetbond. The back skin was cut along the rostral caudal axis, and the

spine was fixed into stereotaxic spinal clamps (Kopf). Muscle and conjunctive tissue were removed before performing a laminectomy along the rostral caudal axis. Spinalization was performed in between the thoracic segments 5 and 6. For electrical stimulations, a 1 MΩ tungsten electrode (World Precision Instruments) was lowered at each microstimulation site (ranging from -550 to -950 μm of depth in the dorsal ventral axis, in each spinal cord segment), and currents ranging from 50 to 140 μA were injected using a stimulus isolator (Model no. A365RC, World Precision Instruments) while possible movements were documented. Various stimulation protocols were tested (single, 5 Hz, 50 Hz). Eventual activity in the BSM and TA was monitored via EMG recordings using a custom-made amplifier (×1000) and filtered at 2 kHz. Data acquisition and analysis were performed using spike2 software (CED Cambridge). Tungsten electrodes were moved along the rostral caudal axis of the spinal cord and electrolytic lesions were placed at the location where the biggest EMG responses were encountered. After having mapped the BSM responses along the rostral caudal axis, consecutive electrical stimulations were performed at the spinal cord sites where prominent BSM responses were encountered previously. Inter-stimulus intervals of consecutive electrical stimulations ranged from 15 to 5 min.

In a subset of animals undergoing electrical stimulations (N = 15 mice), sexual behavior (see below) was performed prior to the acute recordings. Animals either ejaculated (N = 8 mice) or performed 5 mounts with intromission (N = 7 mice), before they were anesthetized.

During control experiments, the animals were prepared as above but a waiting time of 1 h was kept before the first stimulation to control that the observed effects are not due to a rundown of the preparation.

For optogenetic stimulations of the Gal+ cells (N = 13 of Gal-ChR2 male mice) and BSM-MNs (N = 10 of C57BL6 mice infected with an AAV expressing ChR2 on the BSM) the experimental procedure was similar except the fact that an optrode (diameter: 1 mm) was moved on top of the spinal cord, in the rostral caudal axis, while monitoring EMG recordings and documenting movements. Likewise, an electrolytic lesion was placed at the position where the light pulses led to the most prominent BSM EMG responses. We tested a variety of stimulation protocols (single, 5 Hz, 10 Hz, 20 Hz, 50 Hz, 100 Hz) and laser powers (5–40 mW).

**In vivo juxtacellular recordings of photoidentified cells.** Single in vivo juxtacellular recordings were performed as described in ref. 30. Briefly, a glass electrode (resistance ranging from 4 to 6 MΩ) made of borosilicate glass tubes (Hilgenberg) was first lowered at the position where optogenetic stimulations led to the highest activity in the BSM. Pipettes were filled with a ringer solution. Extracellular local field potential recordings were captured while shining the light. Subsequently, cells were searched for by applying a negative current pulse and using an audio monitor (Grass Technologies, AM10) while steps were made in 1.5 μm increments with a micromanipulator (Luigs & Neumann SM-5, Germany) and potential increases in resistance were carefully observed. When spiking activity was detected, electrophysiological recordings were performed in line with optogenetic stimulation protocols (single, 5 Hz, 10 Hz, 20 Hz, 50 Hz) and EMG monitoring. Recordings were amplified (Dagan BVC-700A, Dagan, Minneapolis, MN), low-pass filtered at 10 kHz and sampled at 50 kHz by a data-acquisition interface (Power 1401, CED, Cambridge, England) and controlled and analyzed by the spike2 software (CED, Cambridge, England).

**In vitro patching of identified BSM-MNs in new-born male Gal-ChR2 animals.** New-born male Gal-ChR2 mice aged postnatal day (P2) to P6 were used in accordance with the guidelines of the French Agriculture and Forestry Ministry for handling animals. The protocol was approved by the local ethics committee of the University of Bordeaux

(approval number 2016012716035720). To record specifically from identified MNs innervating the BSM, a crystal of cholera toxin β-subunit conjugated to AlexaFluor 594 (Thermo Fisher Scientific, C34777) was inserted into the BSM with an insect pin 20–24 h before slice preparation procedure in cryo-anesthetized mouse pups. Following the labeling process, spinal cord slices were prepared using the following procedure: mice were anesthetized using isoflurane until all reflexes were gone. After decapitation, the spinal cord was dissected out in an ice-cold sucrose-based saline solution containing the following: 2 mM KCl, 0.5 mM CaCl₂, 7 mM MgCl₂, 1.15 mM NaH₂PO₄, 26 mM NaHCO₃, 11 mM glucose and 205 mM sucrose. The saline was bubbled with 95% O₂, 5% CO₂. Transverse slices (350 μm) of the lower lumbar enlargement and first sacral segments were cut with a vibratome and then transferred to a holding chamber. Slices were allowed to recover in oxygenated aCSF (130 mM NaCl, 3 mM KCl, 2.5 mM CaCl₂, 1.3 mM MgSO₄, 0.58 mM NaH₂PO₄, 25 mM NaHCO₃, 10 mM glucose) for at least 1 h at 30 °C. Whole-cell current-clamp recordings from BSM-MNs, identified by their 594 fluorescence, were made under visual control with a Multiclamp 700B amplifier. Recording glass microelectrodes (4–7 MΩ) were filled with the following: 120 mM K-gluconate, 20 mM KCl, 0.1 mM MgCl₂, 1 mM EGTA, 10 mM HEPES, 0.1 mM CaCl₂, 0.1 mM GTP, 0.2 mM cAMP, 0.1 mM leupeptin, 77 mM d-mannitol and 3 mM Na2-ATP, with a pH of 7.3. All the experiments were performed at room temperature (-23 °C). Data acquisition and analysis were performed using the Axograph software. Experiments were discarded if series resistance increased more than 20% during a given recording period. Polysynaptic transmission was decreased using a high cation solution containing 7.5 mM CaCl₂ and 8 mM MgSO₄ (Liao and Walters, 2002). Throughout recording episodes, GABAergic and glycinergic inputs were blocked with gabazine and strychnine (1 μM each), respectively[88,89]. A stimulating optrode connected to an optogenetic laser box (Prizmatrix) was placed above the central canal at the dorsal gray commissure and light pulses were applied at different lengths. Excitatory postsynaptic currents (EPSCs) were recorded from MNs held at −60 mV in current clamp mode. The input resistance of MNs (Rin) was determined from the slope of the voltage-current curve within the linear portion of current traces. AHP parameters were measured after single action potential evoked by short depolarizing current steps (7 ms, 0.25 nA) in current clamp conditions in MNs held at −60 mV by injection of bias current.

**Sensory stimulation of the penis and local field potential recordings of photo-identified cells**
For sensory stimulation of the penis in parallel with BSM activity monitoring by EMG, the surgical procedure was performed as described above (In vivo *electrical stimulations and optogenetic stimulations*). To stimulate the penis electrically (by wrapping a nerve cuff electrode around the penis) or mechanically (by locally applying an air puff) the penis was gently pulled out. Electrical (6 V) and mechanical (0.5–1 mbar) stimulations of the penis were done at either 200 Hz (100 pulses) or 5 Hz (3 × 5 pulses, 100 ms). Stimulation protocols were first run in an anesthetized male mouse with an intact brain spinal connection before disrupting the latter by performing a spinalization between the thoracic segments 9 and 10. Electrical and/or mechanical stimulation protocols were then repeated in the spinalized preparation (N = 7 mice).

For local field potential (LFP) recordings of optogenetically tagged Gal+ or BSM-MNs, a glass electrode (resistance ranging from 4 to 6 MΩ) made of borosilicate glass tubes (Hilgenberg) was lowered at the position of the Gal+ cells or the BSM-MNs and optogenetic stimulations were performed until encountering the maximal light triggered LFP response. Subsequently, the glass pipette was left in the place at which the highest light triggered LFP response of BSM-MNs or Gal+ cells was encountered. Finally, the penis and leg were mechanically stimulated with a locally applied airpuff (5 Hz; 5 pulses, 100 ms)

while monitoring LFP responses and EMG responses in BSM and TA muscles. Analysis was performed in spike2 and data plotted in Excel and Matlab.

## Behavior
**Sexual priming prior electrical stimulations.** Male mice were single-housed and trained in sexual behavior, with a primed ovariectomized female, until ejaculation was reached in 3 sessions. Afterwards, the animals were divided into two groups: one group was allowed to perform 5 mounts with intromissions ($N = 7$ mice), and the second group was allowed to perform the whole repertoire of sexual behavior until reaching ejaculation ($N = 8$ mice). Immediately after the behavior, the animals were anesthetized and used for in vivo electrical stimulations as described above.

**cFos experiments.** Male Gal-cre x tdTomato mice were single-housed and trained in sexual behavior, with a primed ovariectomized female, until ejaculation was reached in 3 sessions. After one week, the animals underwent a behavioral paradigm and were divided into three different groups: in the first group, animals were allowed to socially interact with a female for 10 min (Aroused, $N = 7$ mice), without performing mounts or intromissions. Note that the mice were prevented from mounting by putting the hand into the experimental arena thereby separating the mice. The second group of male mice performed five mounts with intromissions (5 Mounts, $N = 5$ mice), after which the female was removed and in the third group, the animals were allowed to reach ejaculation (Ejaculation, $N = 6$ mice). In another group of animals (Cage control) male mice were either alone in their home cages, without sexual encounters or female odors present ($N = 3$ mice), or had the same sexual training but were then alone in the clean behaving box for 10 min ($N = 3$ mice), in order to assess the baseline neuronal activity in the spinal cord (no differences in the cFos count was observed in these two control conditions). In the last group, animals ($N = 5$ mice) previously sexually trained, were allowed to interact with another male mouse (5 weeks old) for 10 min or until 10 attack bouts happened. After the behavior session, the experimental mice were placed back in their home cages for 90 min to allow for sufficient cFos expression. Finally, the animals were deeply anesthetized and perfused transcardially with saline, followed by a cold 4% PFA in 0.01 mol/L PBS. Spinal cords were removed from the spine and kept for 1 h in 4% PFA before transferring them for another hour into 0.01 M PBS. Subsequently, spinal cords were stored overnight in 30% sucrose in 0.01 M PBS, 0.1% azide to cryo-protect the tissue. Spinal cords were embedded in frozen section medium for half an hour at −80 °C in 2-methylbutane solution before mounting them in the cryostat. The spinal cords were sectioned at 30μm for post-hoc immunohistochemical staining as described above.

**DTR experiments.** Male Gal-cre x tdTomato animals (2–3 months old) were single-housed and trained for sexual behavior until ejaculation was reached once. After that, the animals were spinally injected with an AA8V-FLEX-DTR-GFP ($N = 12$ mice) at the location of the SEG cells (allowing for the specific expression of the Diphtheria Toxin Receptor) or had a sham surgery ($N = 7$ mice), as described above, and allowed to recover for 2 weeks. Afterwards, the animals underwent another round of sexual behavior training to confirm that they were still reliably ejaculating. After this, the animals received an intraperitoneal injection of 50 ng/g (0.1 mL/10 g) of Diphtheria Toxin (DT, Sigma, D0564-1MG). One week after, the animals were transferred to an experimental box, where they had a 10 min habituation, after which a primed ovariectomized female was introduced. The animals were allowed to perform the full repertoire of sexual behavior and after ejaculation was reached, or 90 min after female introduction passed when sexual behaviors were observed, the animals were returned to their home cage. If the animals did not initiate behavior, the experiment was stopped after 30 min and the sexual experimental paradigm was repeated for two more weeks. In the final session, or after ejaculation, the animals were returned to their home cage and a 90 min interval was kept before perfusion, for cFos expression to occur. Finally, the animals were deeply anesthetized, transcardially perfused and the spinal cords collected for histological processing as described above (*cFos experiments*).

**Erection assessment in DTR experiment.** To assess the effect of the Gal+ cells ablation on the erection of DTR animals, a set up with a glass at the bottom was used to observe the pelvic area movements on DTR and sham animals. The set up consists of a mirror at a 45 degrees angle below the experimental box and a glass floor, allowing for the recording of the side view and bottom view of the sexual encounter with the same camera. Male mice underwent the DTR experiments behavioral protocol explained above and had an additional experimental week where they were transferred for the glass set up and the pelvic movements were recorded (Supplementary videos 3 and 4) while the animal performed sexual behavior until ejaculation was reached. Afterwards, the animals were returned to their home cage and a 90 min interval was kept before perfusion, for cFos expression to occur. Finally, the animals were deeply anesthetized, transcardially perfused and the spinal cords collected for histological processing as described above.

**In vivo BSM EMG recordings.** Male mice were single-housed and trained in sexual behavior, with a primed ovariectomized female, until ejaculation was reached in 3 sessions. Afterwards, the animals were implanted with EMG probes in the BSM as described above. Following a recovery period of 2 weeks post-surgery in their home cages, the animals were put on an experimental box, where they had a 10 min habituation, after which a primed ovariectomized female was introduced. The animals were allowed to perform the full repertoire of sexual behavior until ejaculation was reached, and they were transferred back to their home cage. If the animals did not initiate behavior, the experiment was stopped after 30 min and the sexual experimental paradigm was repeated for at least two more weeks. At the end of the experiment, the animals were sacrificed and the probes recovered to reuse in future experiments. BSM recordings were conducted using an RHD 16-Channel bipolar-input recording headstage (Intan Technologies), connected to an Acquisition Board (Open Ephys), using a sampling rate of 30 KHz. The board and cameras were controlled using a bonsai script (Bonsai Visual Reactive Programming) and the data was band-passed between 850–7,000 Hz and further analyzed using Matlab.

## Ovariectomy and hormonal priming
All female mice ($N = 20$ mice) used as sexual stimuli were ovariectomized. Briefly, female mice were anesthetized using 3% isoflurane in oxygen and placed into a mouthpiece allowing for continuous isoflurane anesthesia. After all reflexes were gone, an incision was made at the center of the lower back. The skin was separated from the muscle towards both sides of the back. After that, on one side, leveled with the hindlimb, the ovary fat pad was located, and a small incision was made in the muscle above this area. The ovary was gently pulled and the connection between the ovary and the uterus was cut using a cauterizer. The same procedure was repeated for the second ovary. Finally, the incision in the back was sutured and the animal was allowed to fully recover on a heating pad. After two weeks of recovery, the animals underwent hormonal priming during which they received an estrogen (1 mg/ml, Sigma E815 in sesame oil) injection 2 days prior to the sexual behavior experiment and a progesterone (5 mg/ml, Sigma P0130 in sesame oil) injection 4 h before the experiment was scheduled. Hormonal priming with estrogen and progesterone was conducted before each experiment.

## Quantification and statistical analysis

**Histological analysis.** After immunohistochemical labeling of cFos (as described above), the spinal cord sections were imaged using a Slide Scanner (Zeiss AxioScan.Z1, Zeiss Microscopy). The obtained images were processed using the Zen Software (Zen 2.6, Zeiss Microscopy), and tiff images of each channel (488 for the cFos, cy3 for the Gal-tdT+ cells and cy5 for the Nissl staining) were exported. These tiff files were, together with the corresponding image of the spinal segment from the mouse spinal cord atlas[90], opened in Photoshop (Adobe Photoshop, Adobe). Once the overlap between the spinal atlas outlines and the immunohistochemical tiff image was adjusted, the cells present in laminae X (around the central canal and the location of Gal+ cells) of the spinal cord were manually counted by placing dots on each cell. This counting procedure was done for every other slide given the thickness of the slices and thus to prevent double counting. The manual counting procedure was done for both channels, cFos in green and tdtomato in red, by painting circles in Photoshop over the manually scored elements. The Matlab function *imfindcircles* was used to detect and quantify these marks. To identify overlap between the two channels, the Matlab function *knnsearch* was used. If two centroids were closer than the radius of a soma, they were considered as co-localized elements. Manual counting of cFos and Gal-tdT+ cells was done blindly to prevent bias. Different experimenters were involved in the counting: ARM counted the animals for the cFos experiment, except for 3 cage alone animals that were counted by CL; LF counted the animals for the Diphtheria Toxin experiment. For the cell normalization presented in Fig. 6, we divided the number of cFos-positive and Gal-tdT+ cells by the total amount of Gal-tdT+ cells.

**Behavioral analysis.** The DTR, cFos and in vivo BSM recordings behavioral experiments were recorded using two-point gray cameras (Teledyne FLIR), at 60 frames per second. The cameras acquired a top and front video of the cage and were controlled using a bonsai script (Bonsai Visual Reactive Programming). Afterwards, the videos were analyzed using Python Video Annotator (developed at the Champalimaud Foundation). Annotations were conducted in a blind fashion. A wide range of behaviors was annotated, namely: sniffing of the anogenital area of the female; mount attempts (when the male was not able to perform intromissions) mounts with probing (shallow pelvic movements when the male is trying to intromit but still has not inserted the penis); mounts with intra-vaginal thrustings; intra-vaginal thurstings (when the male successfully inserted the penis in the female's vagina); and finally ejaculation (time between the beginning of shivering and the moment the male dismounted the female). The duration, in frames, for each behavior was then aligned to the camera timestamps, acquired during video recording. Analysis of behavioral timestamp data was performed using a python script in Spyder 3.3.6 (Python).

## Statistical analysis

Statistical analysis was performed in Matlab (*ttest2, anova2*) and Python (scipy and statsmodels). All error ranges represent standard error of the mean. For two-sample comparisons of a single variable, Student's t test was used, unless in cases when the underlying distributions were non-Gaussian (Shapiro-Wilk test, $p < 0.05$), where a two-tailed Mann-Whitney-U Test was performed. When multiple variables were compared, a Kruskal Wallis test or a Wilcoxon-Signed-Rank test were used since the data did not follow a Gaussian distribution. Probabilities of the null hypothesis $p < 0.05$ were judged to be statistically significant. Elements of violin plots: center line, median; box limits, upper (75) and lower (25) quartiles; and whiskers, 1.5x interquartile range. Linear regression was performed in Python using *seaborn.regplot*, the correlation coefficient and *p*-value were computed with *scipy.stats.pearsonr*.

The distribution of Gal-tdT+ cFos+ cells along the L2/L3 spinal segments (Fig. 7G) was performed in Julia (with the KernelDensity package). For each animal, a kernel density estimate of the cell counts along the L2/L3 spinal segments was computed. Then we reported the bootstrapped median and interquartile range (IQR) from 1000 kernel density estimates drawn with replacement from all the animals per condition.

To estimate the inter-thrust interval (ITI) versus mount progression (Fig. 7T), the thrust number was divided by the total number of thrusts in the mount, what we call mount progression. Then data from all the mounts and all the sessions from the different groups were pulled together and a non-parametric kernel regression was performed. The shadow region corresponds to the standard deviation.

A power analysis test was done for the cFos experiment due to the lower number of animals used (5–10 per test group). The G Power software was used with the following parameters: test family—F tests; statistical test—ANOVA: Fixed effects, omnibus, one-way; type of power analysis—Post hoc: Compute achieved power—given α, sample size, and effect size; effect size f—1,457217 (calculated using the means of 0,09082 (cage control), 0,1617 (sexually aroused), 0,1523 (male interaction), 0,1800 (5 mounts), 0,3613 (ejaculation), respective group sizes of 8, 10, 5, 5, 6 and a standard deviation of 0,06092); α error probability—0.05; total sample size—34; number of groups—5. This resulted in a power of 99%, with a noncentrality parameter λ of 6,2500000, a critical F of 2,4674936, a numerator df of 4 and a denominator df of 95.

## Reporting summary

Further information on research design is available in the Nature Portfolio Reporting Summary linked to this article.

## Data availability

All data reported in the current manuscript can be found in the following Mendeley data repository https://doi.org/10.17632/xyh6wwd7fz.1. Any further request can be addressed to the corresponding authors (C.L. and S.Q.L.). Source data are provided with this paper.

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

## Acknowledgements

We thank B. Roska for reagents (PRV) and much valuable advice, and A. Sathyamurthy for technical assistance (immunohistochemistry protocol for cFos in the spinal cord). We thank the Lima lab for their critical input during this project. We also thank the community at the Champalimaud Research, in particular the Carey Lab for reagents and technical assistance, the ABBE Facility, the Scientific Hardware, Histopathology and Rodent Champalimaud Research Platforms for technical assistances, and Zachary Mainen for reviewing the manuscript. We thank Diana Costa for the figure illustrations used throughout our manuscript. This work was developed with the support from the Champalimaud Foundation, the Institut de Neurosciences Cognitives et Intégratives d'Aquitaine (INCIA), a Human Frontier Science Program Postdoctoral Fellowship (LT000353/2018-L4) (C.L.), a H2020 Marie Skłodowska-Curie Actions Individual Fellowship (799973) (C.L.), a Fundação para a Ciência e a Tecnologia (FCT) PhD Fellowship (PD/BD141576/2018) (A.R.M.), the research infrastructure Congento, cofinanced by the Lisboa Regional Operational Programme (Lisboa2020), under the PORTUGAL 2020 Partnership Agreement, through the European Regional Development Fund (ERDF), FCT under the project LISBOA-01-0145-FEDER-022170, an InterEmerging Actions 2020 (301137) (S.Q.L.) and an European Research Council (ERC) Consolidator Grant (772827) (S.Q.L.). We thank the National Institutes of

Health (NIH) for their support with the Myomatrix EMG electrodes (NIH grants U24NS126936 and R01NS109237).

## Author contributions

C.L., A.R.M., L.F., H.M., C.Q., and S.B. performed experiments. C.L., A.R.M., N.G.C., and B.L. performed all data analysis. C.L. and S.Q.L. supervised the study. C.L., A.R.M., and S.Q.L. wrote the manuscript. S.B. reviewed and edited the final version of the manuscript.

## Competing interests

The authors declare no competing interests.
