## [Transparent Peer Review file · Nature Communications]

A galanin-positive population of lumbar spinal cord neurons modulates sexual arousal and copulatory behavior in male mice

Corresponding Author: Dr Susana Lima

Version 0:

Reviewer comments:

Reviewer #1

(Remarks to the Author)

I reviewed this manuscript earlier. I think the authors have done a good job addressing my concerns.

Reviewer #2

(Remarks to the Author)

The revised manuscript shows substantial improvement, with many ambiguous points now clarified. I appreciate the authors' efforts to improve the work by adding technically challenging experiments. The in vivo BSM EMG recording during sexual behavior is especially impressive. The manuscript is now suitable for publication, though I have some minor suggestions and comments for further clarity:

- Fig1M legend: Please add "MN" after "~ single BSM-MNs (".
- Fig2 B and C: Please indicate which lambar sections are shown in the images.
- Fig2E "Gal/td+cells" is confusing. Please replace it with "Gal-tdT+ cells" instead. Please maintain this nomenclature throughout all texts and figures (including Figure 7).
- Throughout the paper, I recommend using "Gal-tdT" neurons when the neurons were defined as Gal+ by using transgenic animals, not with antibody staining.
- Suppl Fig6: Please indicate that green "Galanin" signals are from immunohistochemistry in the legend. Please indicate that the image in panel C is duplicated from Fig. 2C.
- Suppl Fig13G: missing legend.
- There appears to be a discrepancy between FG+ and PRV+ neuron distributions. Fig1C and Suppl Fig1 show minimal or no FG+ cells in L2/3, while Suppl Fig9 shows abundant PRV+ neurons in the upper spinal cord. Could the authors explain this difference?
- Regarding Page9: "These results suggest that the areas sending information to the Gal+ cells receive reciprocal input and ~~~" and Page24: "Moreover, we provide strong anatomical evidence for a reciprocal connection between the Gal+ neurons and autonomic nuclei, as well as the ischiocavernous MNs."
If I understand correctly, Suppl Fig5 indicates Gal+ terminals were found in IMK, CAN, SNB, SPN, and DLN, meaning these areas receive inputs from Gal+ neurons. Suppl Fig6 shows that all these areas, except DLN, have PRV+ neurons, meaning these areas project to BSM.
The term "Receiving reciprocal input" suggests IMK, CAN, and SNB both receive projections from Gal+ neurons (data

shown) and project to Gal+ neurons. Is there literature that shows these areas project to Gal+ neurons?

- (This is just a comment) As the authors mentioned in the rebuttal, Gal immunohistochemistry signals are dotted, making it difficult to identify the cell bodies. This might be due to the fast axonal transport of Gal peptide. Colchicine treatment to inhibit axonal transport (often used in the staining of neuropeptidergic neurons) might help improve staining. (eg. Kondo et al., Cell Tissue Res (1988) 251:221-224, Ribak et al., Brain Research, 140 (1978) 315-332) Alternatively, in situ hybridization may work better.

We thank the referees for the positive assessment of our work. Below, we provide point-by-point responses to each of the reviewers' comments (in italic), addressing their concerns and incorporating their suggestions (which are colour highlighted in the manuscript).

Reviewer #1

I reviewed this manuscript earlier. I think the authors have done a good job addressing my concerns.

We thank the reviewer for the positive feedback.

Reviewer #2

The revised manuscript shows substantial improvement, with many ambiguous points now clarified. I appreciate the authors' efforts to improve the work by adding technically challenging experiments. The in vivo BSM EMG recording during sexual behavior is especially impressive. The manuscript is now suitable for publication, though I have some minor suggestions and comments for further clarity.

We thank the reviewer for the positive assessment of our work and the kind words regarding the last set of experiments added to the manuscript.

1) Fig1M legend: Please add "MN" after "~ single BSM-MNs".

We thank the reviewer for pointing this out, we have changed the figure legend according to the suggestion.

2) Fig2 B and C: Please indicate which lumbar sections are shown in the images.

We thank the reviewer for pointing this out, we have included the lumbar sections of the images in the two panels from Fig2.

3) Fig2E "Gal^{td}+cells" is confusing. Please replace it with "Gal-tdT+ cells" instead. Please maintain this nomenclature throughout all texts and figures (including Figure 7).

We thank the reviewer for pointing this out, we have replaced "Gal^{td}+cells" with "Gal-tdT+ cells" throughout the whole manuscript.

4) Throughout the paper, I recommend using "Gal-tdT" neurons when the neurons were defined as Gal⁺ by using transgenic animals, not with antibody staining.

We thank the reviewer for pointing this out, we changed the manuscript to the recommended nomenclature.

5) Suppl Fig6: Please indicate that green “Galanin” signals are from immunohistochemistry in the legend. Please indicate that the image in panel C is duplicated from Fig. 2C.

We thank the reviewer for pointing this out, we incorporated all changes in the manuscript.

6) Suppl Fig13G: missing legend.

We thank the reviewer for pointing this out, we added the legend to Suppl Fig13G.

7) There appears to be a discrepancy between FG+ and PRV+ neuron distributions. Fig1C and Suppl Fig1 show minimal or no FG+ cells in L2/3, while Suppl Fig9 shows abundant PRV+ neurons in the upper spinal cord. Could the authors explain this difference?

We thank the reviewer for the comment, however we believe we cannot compare the two experiments directly, for two main reasons. While FG only labels MN that were retrogradely labeled after injecting the dye in the muscle (first order neurons), the PRV will retrogradely label MN (similar to FG), but will also label presynaptic neurons connecting to the MNs (second order neurons). This means that the cells labeled with PRV in L2/3 are a mix of first order and second order neurons, which we cannot distinguish as they all show up as green. In addition, viral particles spread much more than FG, which means that the number of infected cells is lower with FG. These two factors are contributing to the lower number of cells.

8) Regarding Page9: “These results suggest that the areas sending information to the Gal+ cells receive reciprocal input and ~~~” and Page24: “Moreover, we provide strong anatomical evidence for a reciprocal connection between the Gal+ neurons and autonomic nuclei, as well as the ischiocavernous MNs.”

If I understand correctly, Suppl Fig5 indicates Gal+ terminals were found in IMK, CAN, SNB, SPN, and DLN, meaning these areas receive inputs from Gal+ neurons. Suppl Fig6 shows that all these areas, except DLN, have PRV+ neurons, meaning these areas project to BSM. The term “Receiving reciprocal input” suggests IMK, CAN, and SNB both receive projections from Gal+ neurons (data shown) and project to Gal+ neurons. Is there literature that shows these areas project to Gal+ neurons?

Through our synaptophysin experiment we have shown that these nuclei receive inputs from the Gal+ cells (Supplemental Figure 5 clearly shows green-positive boutons in the above mentioned areas). Furthermore, our PRV experiment, which labels neurons retrogradely, indicates that these nuclei are either sending projections to the Gal+ cells or directly to the BSM-MNs, since the only way to become a PRV-positive neuron is if the virus jumps from the BSM-MNS retrogradely to the Gal+ cells and again retrogradely to this nuclei (Supplemental Figure 6). Which points for an input from the IMK, CAN, and SNB to the Gal+ cells and therefore a reciprocal connection between these nuclei and the Gal+ cells. Of course this is just based on anatomical and not functional evidence, so further experiments are needed to further establish these connections.

Regarding previous literature, to our knowledge in the mouse there is no anatomical study that shows these reciprocal connections, so our study would be the first to notice this pattern. However, Xu et al, for example, have described this connection between the autonomic nervous system and the Gal+ cells in the rat after PRV injection in the BSM.

*Xu, C., Giuliano, F., Yaici, E. D., Conrath, M., Trassard, O., Benoit, G., & Vergé, D. (2006). Identification of lumbar spinal neurons controlling simultaneously the prostate and the bulbospongiosus muscles in the rat. *Neuroscience*, 138(2), 561–573. <https://doi.org/10.1016/j.neuroscience.2005.11.016>*

- (This is just a comment) As the authors mentioned in the rebuttal, Gal immunohistochemistry signals are dotted, making it difficult to identify the cell bodies. This might be due to the fast axonal transport of Gal peptide. Colchicine treatment to inhibit axonal transport (often used in the staining of neuropeptidergic neurons) might help improve staining. (eg. Kondo et al., *Cell Tissue Res* (1988) 251:221-224, Ribak et al., *Brain Research*, 140 (1978) 315-332) Alternatively, in situ hybridization may work better.

We thank the reviewer for the comment which we will take into consideration for future experiments.